# Asynchronous domain dynamics and equilibration in layered oxide battery cathode

Zhichen Xue[1,2], Nikhil Sharma [3], Feixiang Wu [1] ✉, Piero Pianetta [2], Feng Lin [4], Luxi Li [5] ✉, Kejie Zhao [3] ✉ & Yijin Liu [6] ✉

To improve lithium-ion battery technology, it is essential to probe and comprehend the microscopic dynamic processes that occur in a real-world composite electrode under operating conditions. The primary and secondary particles are the structural building blocks of battery cathode electrodes. Their dynamic inconsistency has profound but not well-understood impacts. In this research, we combine operando coherent multi-crystal diffraction and optical microscopy to examine the chemical dynamics in local domains of layered oxide cathode. Our results not only pinpoint the asynchronicity of the lithium (de)intercalation at the sub-particle level, but also reveal sophisticated diffusion kinetics and reaction patterns, involving various localized processes, e.g., chemical onset, reaction front propagation, domains equilibration, particle deformation and motion. These observations shed new lights onto the activation and degradation mechanisms of state-of-the-art battery cathode materials.

Thanks to their robust lattice configuration and superior chemical stability, lithium layered oxide materials, e.g., $LiNi_xMn_yCo_zO_2$ (NMC), $x + y + z = 1$, have been broadly adopted as cathodes for high-energy-density lithium-ion batteries. From the thermodynamic perspective, the energy storage and release in lithium-ion batteries are executed through the reversible (de)intercalation of lithium ions and the associated redox reactions in the hosting lattice matrix[1]. In practical implementations, however, lithium-ion batteries are exceedingly complex systems, and the charge and discharge behavior is greatly affected by several kinetic processes, e.g., ion hopping in the crystal lattice[2], ion transport at the interface[3], solvation/de-solvation[4], electron conduction[5], and ion diffusion in the liquid electrolyte[6]. These processes collectively affect the reaction dynamics over a wide range of spatial and temporal scales, posing significant practical impacts on the electrochemical performance at the cell level. To meet the growing demands in the rapidly transforming industries of consumer electronics, electric vehicles, and long-duration energy storage systems, it is important to conduct a deep dive into these fundamental reaction mechanisms.

Conventional diagnostic methods in this field have placed a strong emphasis on the electrochemistry and the bulk-averaged material properties. For example, a suite of electrochemical testing protocols including cyclic voltammetry (CV), $dQ/dV$, electrochemical impedance spectroscopy (EIS), and galvanostatic intermittent titration technique (GITT) have been developed to evaluate the cell performance and to interpret the electrochemical behavior[7,8]. These cell-level tests can be used to infer a range of chemical and kinetic processes, but they do not have direct sensitivity to the microscopic characteristics and, thus, rendering an averaged effect. For battery material characterizations, scattering[9], spectroscopy[10], and microscopy methods with X-ray[11,12], neutron[13], electron[14,15], and optical probes[16–18] have been broadly applied. Although these characterization tools can probe a

[1]School of Metallurgy and Environment, Central South University, 410083 Changsha, China. [2]Stanford Synchrotron Radiation Lightsource, SLAC National Accelerator Laboratory, Menlo Park, CA 94025, USA. [3]School of Mechanical Engineering, Purdue University, West Lafayette, IN 47906, USA. [4]Department of Chemistry, Virginia Tech, Blacksburg, VA 24061, USA. [5]X-ray Science Division, Argonne National Laboratory, Lemont, IL 60439, USA. [6]Walker Department of Mechanical Engineering, The University of Texas at Austin, Austin, TX 78712, USA. ✉e-mail: feixiang.wu@csu.edu.cn; luxili@anl.gov; kjzhao@purdue.edu; liuyijin@utexas.edu

range of material properties, including lattice structure, electronic structure, and micromorphology, a direct observation of the microscopic dynamic chemical processes in an operating lithium-ion cell is still challenging.

The microscopic dynamic processes that occur in real-world, industry-relevant layered oxide cathodes are regarded as important factors but have not been extensively studied. Mechanistic understandings in this area can critically inform the efforts in designing and optimizing next-generation lithium-ion batteries. For example, through an in situ X-ray diffraction study of polycrystalline layered oxide cathodes, Park et al. reported a phenomenon that appears to be a thermodynamically forbidden phase separation during delithiation[19]. Their analysis suggests that this fictitious phase separation is caused by heterogeneously distributed local current density and particle reaction. Our previous experimental work using x-ray phase contrast holotomography to statistically quantify thousands of active NMC cathode particles has led to the development of a network evolution model[20,21], which reveals not only the interplay between the activity and degradation of individual particles, but also the mutual regulation among them. These results helped us to understand the underlying mechanism of the observed transition from asynchronous particle activities to synchronous cluster/electrode behaviors as the electrochemical cycling progresses[21]. In addition to the chemical heterogeneity at the electrode scale, state of charge (SOC) inhomogeneity has been observed within individual active cathode particles using high-resolution X-ray and electron microscopy techniques[22,23]. Despite tremendous efforts devoted to this field, it remains a challenge to reveal the microscopic dynamic processes in operating lithium-ion batteries with high temporal resolution and low beam damage.

In this work, we elucidate the domain dynamics in layered cathode by investigating lithium-ion cells using coherent multi-crystal diffraction (CMCD) and optical microscopy. In contrast to conventional bulk-averaged X-ray diffraction approaches, we focus on the dynamic evolutions of the lattice structure in many of the spatially separated primary grains. The CMCD results show that, during battery charging, the NMC domains demonstrate significant differences in their electrochemical activation, local SOC, and physical movement. To corroborate the CMCD observations in the reciprocal space, we employ operando optical microscopy to directly visualize, in real space, an exposed cathode cross-section. The widely conceived shrinking-core model is invalidated in our observation. Instead, it clearly shows a near-surface point onset, followed by a progressive reaction front propagation. This observation holds true for nearly a hundred particles imaged in our experiment, demonstrating a strong statistical significance. It highlights the importance of individual particles' activation energy barrier that one must overcome to initiate the intercalation process. Interestingly, for most of the particles investigated, the amount of time it takes for a particle or a local domain to reach to its maximum SOC from its electrochemical onset is very similar, regardless of the particle/domain size and shape. An in-depth analysis of selected particles demonstrates an intra-particle domain equilibration effect, which features the electrochemical interaction among the spatially adjacent domains. The observation of these interesting phenomena has profound implications for the understanding of capacity loss during the cell activation process and the relationship between micromorphology and chemomechanical processes. Specifically, our results indicate the importance of structural engineering from the particle to electrode levels to accommodate the ubiquitous asynchronous domain dynamics and equilibration in operating batteries.

## Results
### Tracing the domain dynamics in layered oxide cathode with CMCD
During battery operation, the transition metal interlayer spacing in the layered oxide cathode periodically expands and contracts as lithium ions are inserted and extracted. The interlayer spacing of these crystals expands due to the shielding effect of lithium on the O-O repulsion during the delithiation process, and then shrinks with the interlayer slip[24]. Operando X-ray diffraction has been broadly applied to follow the evolution of the lattice parameters as a function of SOC. In conventional powder diffraction, X-ray with a large beam size is used to illuminate the electrode, covering a large footprint, and yielding a ring-shaped diffraction pattern that is reduced to a one-dimensional plot for further analysis. Therefore, although XRD is used to interpret the material's lattice configurations, it is a bulk-averaged signal and cannot track the lattice dynamics within individual grains. To overcome this limitation, we adopt a new approach termed CMCD, which uses a focused coherent beam of X-rays and observes only a small portion of the entire Debye-Scherrer ring at the reflection of interest (as shown in Fig. 1a)[25]. The niche is to achieve a reasonable ratio between the X-ray spot size and the crystal grain size such that the powder "ring" consists of many yet clearly separatable and traceable diffraction spots. Due to this feature, CMCD offers the ability to resolve the microscopic dynamics within and among primary crystal grains in the cathode. As shown in Supplementary Movie 1, the CMCD diffraction patterns clearly demonstrate a set of bright diffraction spots, which correspond to reflections from the (003) crystal plane of dozens of primary cathode grains. Overall, these spots repeatedly move up and down, following the trend of the bulk averaged XRD signal (Fig. 1B).

A closer look at the CMCD patterns, however, reveals the inconsistency among different primary grains. As shown in Fig. 1C, two consecutive diffraction patterns are fused into a color-coded map, in which the red spots are from the current time stamp ($T_i$) and the blue spots are from the previous time stamp ($T_{i-1}$). Several selected regions of interest are enlarged for better visualization and a few interesting processes become evident: (1) the electrochemical onset for different primary grains can occur at different time (Fig. 1d versus Fig. 1f); (2) at a given time-stamp, different primary grains can exhibit different SOC (Fig. 1f); (3) primary grains can experience different electrochemical reaction rates, which could lead to inter-grain heterogeneity and/or equilibration (Fig. 1h); (4) many of the primary grains demonstrate physical movements and rotations in different directions (Fig. 1e, g), indictive of particle and electrode deformation. These observations not only reveal the kinetic inconsistency of ion diffusion in primary NMC domains, but also suggest that the asynchronous primary grain activity can be an overlooked mechanism for the crack formation in NMC secondary particles.

### Visualizing the domain dynamics using operando optical microscopy
To corroborate the CMCD observations in the reciprocal space, we employ optical microscopy to directly image an exposed cathode cross-section during battery cycling. The use of optical microscopy is a low-cost, readily available, low-beam damage approach with chemical and dynamic sensitivities to the electrochemical reactions in layered oxide cathodes. The experimental details and analysis of the inter-secondary-particle behaviors have been reported by some of our co-authors[26]. It has been demonstrated that the optical intensity is monotonically correlated with the SOC[16–18,27], which is consistence with our observations. Therefore, we utilize the normalized optical intensity of the NMC particles as a proxy for the local SOC, and the derivative of which is interpreted as the local current density.

A notable feature of the first charge is that the activation of the cathode particles takes place consecutively, which has been extensively discussed in the earlier publication[26]. In this work, with identification and segmentation of nearly 100 particles, we investigate the microscopic dynamics with statistical significance. It is interesting to point out that, regardless of the size and shape of the particles, it takes around the same amount of time for an individual particle to reach its maximum SOC after its electrochemical onset (Figure S1), while the

entire electrode takes ~15 h to complete the first charge in our experiment. This is quite interesting and motivates an in-depth analysis at the sub-particle level.

To understand the intra-particle dynamics, we select a typical particle for a detailed investigation (Fig. 2). As shown in Fig. 2a, when this particle is actively charged, it gradually increases its optical intensity. By calculating the differential maps using the images with consecutive time stamps, the spatial distribution of the local currents can be visualized. It is interesting that the broadly conceived shrinking-core model is not observed in our experiment. Instead, the particle undergoes a near-surface point onset, followed by a gradual reaction front propagation through the entire particle (Fig. 2b). This holds true for most of the nearly 100 particles imaged in our experiment, featuring strong statistical representativeness. In Fig. 2c, we further plotted the time-dependent intensity evolution for several randomly selected pixels, which clearly demonstrate the asynchronicity of these local regions. The averaged behavior of the entire particle is plotted in black in Fig. 2c, and its derivative, the local current variation over time, is shown in Fig. 2d, indicating that this particle takes ~8 min to

complete its charging process after its electrochemical onset. This anisotropic delithiation pattern could potentially be attributed to the irregularity in the particle shape[28], non-uniform carbon-binder contact[20,29,30], grain structure[31], compositional variations[32] or electrode heterogeneities[33,34], which are often purposely tuned to adjust the cell behavior.

With high-throughput analysis of nearly 100 particles, we single out an interesting one that demonstrates a distinct dynamic behavior. This particle took significantly longer to be fully charged (Fig. 3a). Its optical intensity evolution features a relatively rapid raise near the beginning, which is followed by a slow and steady increase throughout the entire charging process. We enlarge the time window of 20 to 80 mins for a closer look and, interestingly, observe three steps in this region (Fig. 3b, gray). The corresponding current density plot (Fig. 3b, orange) suggests that there are three charging pulses in this time window, each goes into a different local domain (D1, D2, and D3, respectively). These three charging pulses last for 7, 8, and 11 min, respectively. We further extract the normalized intensity profiles for all the four sub-particle domains separately (colored curves in Fig. 3c).

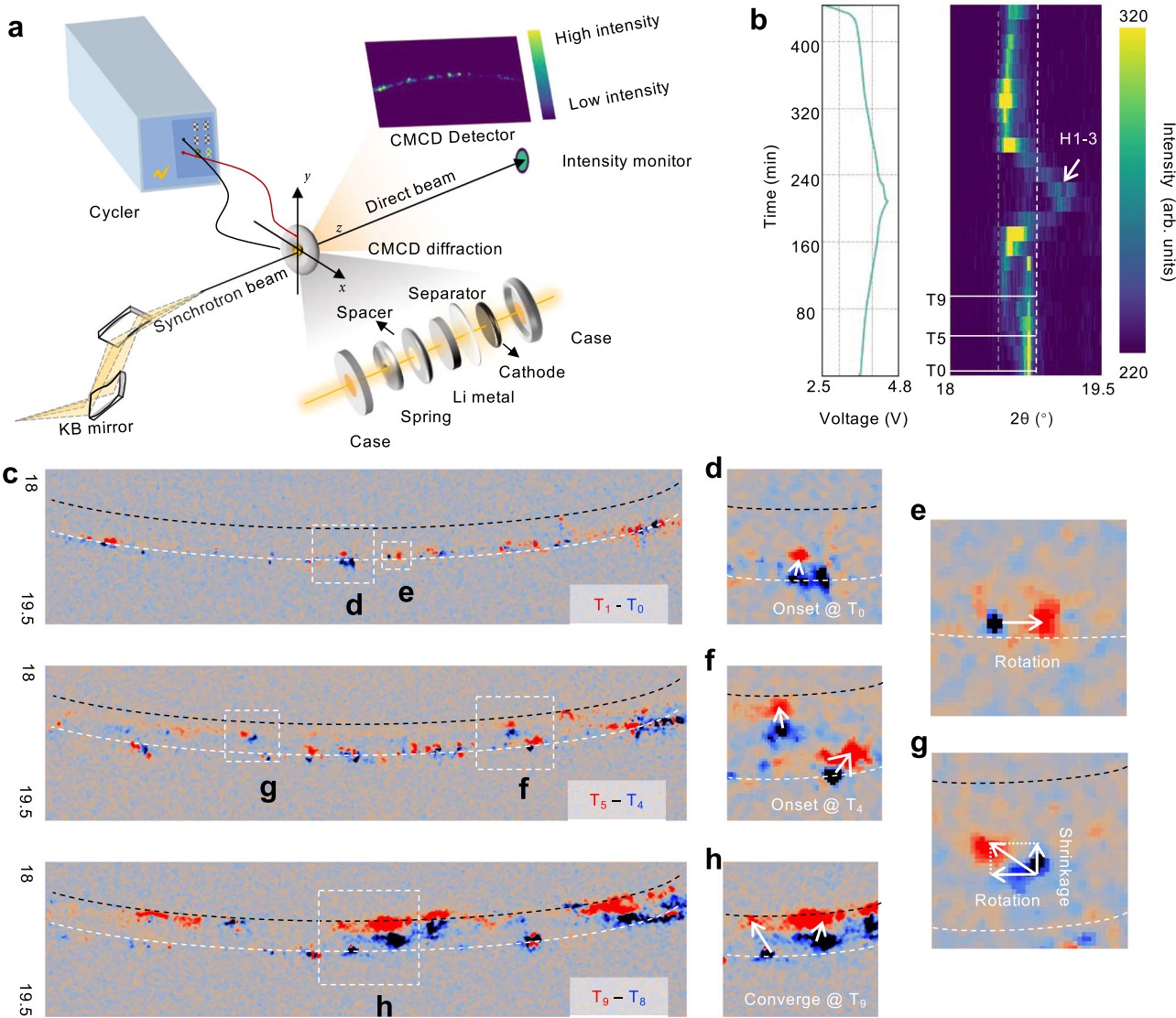

**Fig. 1 | Primary domain dynamics in layered oxide cathode revealed by CMCD.** **a** Schematic of the CMCD technique used in this work. **b** Operando powder X-ray diffraction of NMC cathode's (003) diffraction peak and the corresponding charge-discharge curve of the cell. **c**–**g** Operando CMCD patterns of the (003) Debye-Scherrer ring at several selected time stamps. These maps are differentially fused using two sets of consecutively acquired patterns. The red spots are from the current time stamps, and the blue spots are from the previous time stamps. Time span between two adjacent time steps is 10 min.

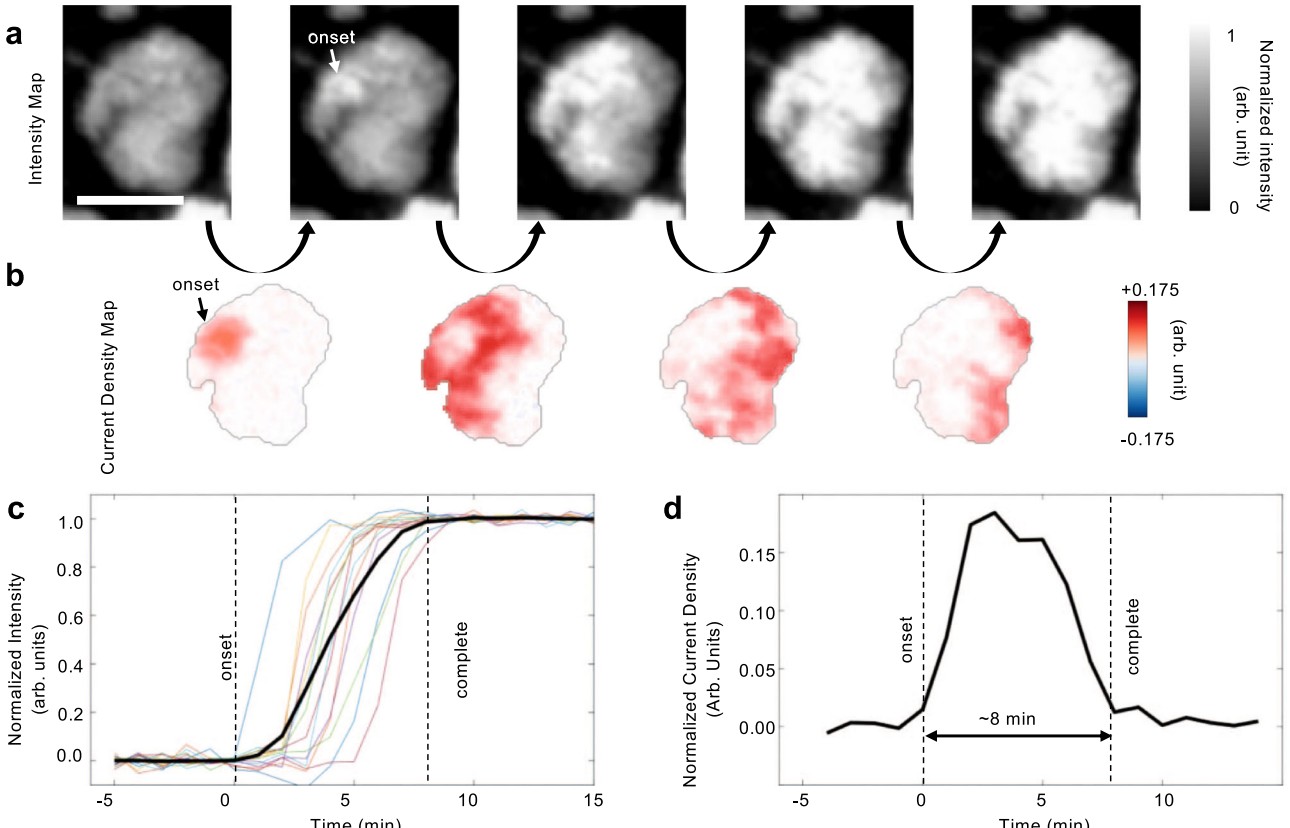

**Fig. 2 | Visualization and quantification of the lithium-ion diffusion front propagation within an NMC particle. a** Images of a typical NMC particle acquired using an optical microscope during the first charge. **b** Sequential differential maps reveal the sub-particle active regions with high local current. **c** Normalized intensity evolution of several randomly selected pixels in the particle (colored) and that averaged over the entire particle (black). **d** The evolution of the current flow over the entire particle. Figures (**a**) and (**b**) share the scale bar which is 10 μm.

Based on this analysis, we identify several critical time stamps for this particle, namely at 38 min, 45 min, 54 min, 65 min, and the end of charge, 891 min. The particle's electrochemical activity maps (Fig. 3d–g, see Methods section in supplementary materials for more details) highlight four different local regions that are being actively charged by different charging pulses. The respective behaviors of domains D1, D2, and D3 are very similar to that of a normal individual particle. Domain 4 (D4), on the other hand, shows a gradual charging behavior over the course of ~15 h, which is quite rare during the first charge. The varying spatial distribution of the local current over this particle is further highlighted by the circular plot in Fig. 3d–g, which demonstrates the sequential activities in the four domains.

To gain a mechanistic understanding of the observed intra-particle asynchronous domain activity, we conducted finite element analysis using a strongly coupled multiphysics theoretical framework[26]. Detailed information regarding the Finite Element Modeling (FEM) model and material properties can be found in the Methods section. Figure 3h–j depicts that the interaction between electrochemistry and mechanical damage controls the intra-particle charging behavior (see Supplementary Movie 2 for more details). The seven domains inside this NMC particle represent intra-particle regions that exhibit chemical homogeneity at the initial state, with directionally isotropic transport and mechanical properties (elastic stiffness and strain). At the particle surface, where the porous carbon binder partially covers the particle, interfacial reactions based on the Butler-Volmer equation occur. The interfacial reaction occurs only at the carbon binder and NMC interface, and lithium diffusion across the domain is controlled through Fick's diffusion law. During charging, as lithium-ions deintercalate the host lattice, NMC domains experience

net lattice volumetric reduction. Consequently, the shrinking of domains generates stress at the domain boundaries, and mechanical detachment/damage occurs. As shown in Fig. 3h to j, the developed heterogeneous nature of the domains within the particle leads to the creation of non-uniform charging behavior among different regions. This asynchronous and heterogeneous charging behavior triggers lattice mismatch, which results in the progressive development of damage. These damages continually restrict and redirect the flow of lithium-ion within the particle, thereby affecting the charging behavior. In Fig. 3k, we observe a monotonic decrease in net particle lithium concentration, with considerable differences in lithium concentration profiles among all seven domains at different time points. These asynchronous reactions are accompanied by an increase in mechanical damage (Fig. S2). The simulation results clearly demonstrate the dynamic nature of the lithium-ion diffusion pathways, which are modulated by intra-particle damage and heterogeneous interfacial reaction with carbon binder. In other words, the relative difference in electrochemical and mechanical properties among domains and the degree of incomplete carbon binder coverage of NMC particles can influence the intensity of such heterogenous intra-particle behavior in composite electrodes. This heterogeneity also leads to further damage of the NMC particles.

We declare that the presented simplistic 2D model does not entirely replicate the experimental configuration. In the real world, the unique sub-particle level dynamic features would be more complicated, involving cracks, defects, impurities, and compositional heterogeneity[35]. Understanding the origin of these heterogeneities is crucial for improving the performance and reliability of batteries. The asynchronicity discussed in this work stems from the electrochemical

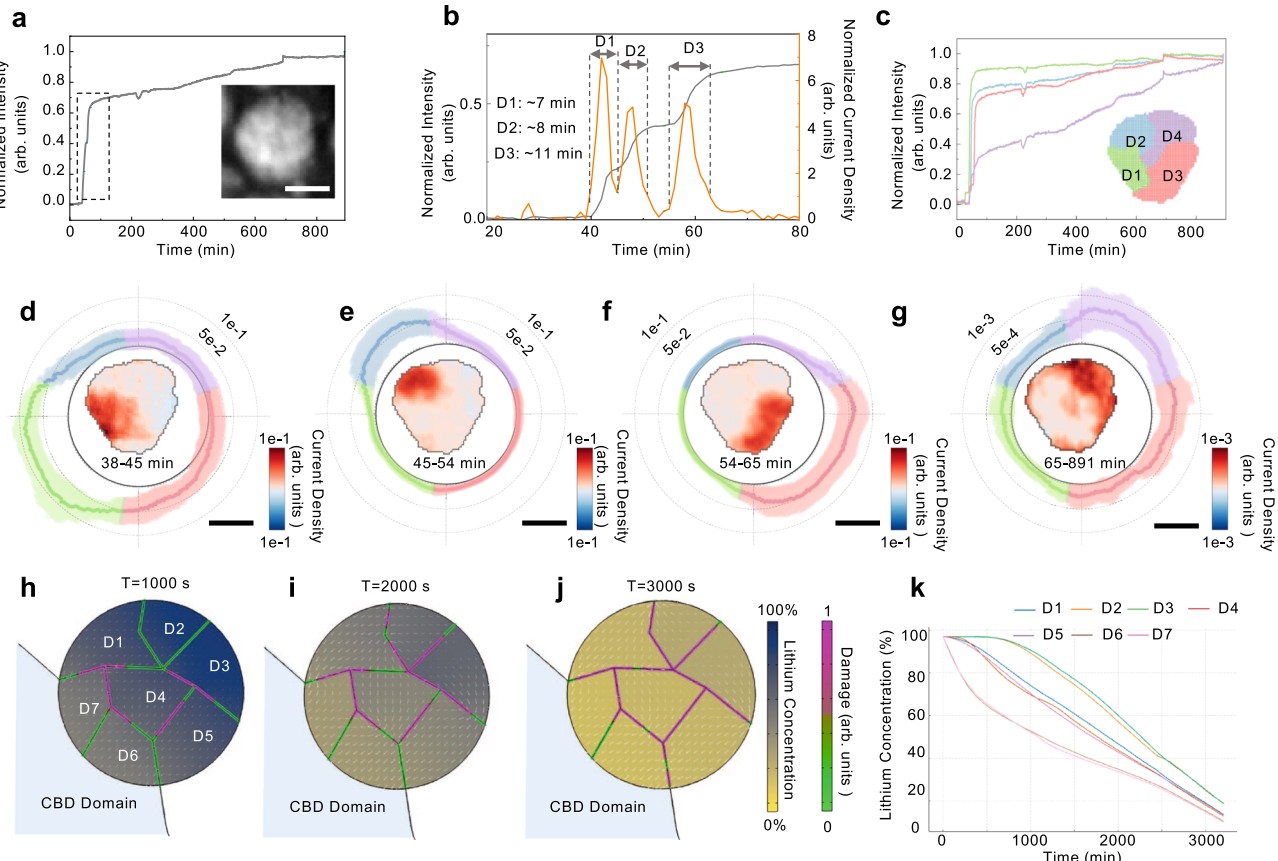

**Fig. 3 | Heterogeneous domain activity within an NMC particle. a** Optical intensity evolution of an abnormal particle upon charging. **b** Optical intensity and current density evolution of this particle in the time window of 20-80 min. Three charging pulses for domains D1, D2, and D3 are identified. **c** The normalized optical intensity evolution of the four different sub-particle domains (D1 to D4). **d–g** the particle's electrochemical activity maps over different time windows and the corresponding circular plot of the local current distributions with error bar (shading).

**h–j** Lithium concentration profiles from finite element analysis of an NMC particle comprising intra-particle domains. The domain boundary experiences damage as NMC particle is charged, thus restricting the lithium flow across them. **k** Lithium concentration profiles for the sub-particle domains evolve over time during the charging process. The domains exhibit an extensive level of intra-domain charge and damage heterogeneity. The scale bars are 5 μm.

kinetics condition heterogeneity and potentially accumulate and escalate the heterogeneity. The identification of distinct sub-particle level dynamic features suggests that battery performance and reliability may be improved by tailoring the internal structure and composition of cathode particles to reduce heterogeneity in electrochemical reactions[36,37].

The co-existence of several local domains with very different electrochemical behaviors within an individual secondary particle echoes our CMCD observations of the asynchronicity in microscopic domain dynamics. Their co-evolution could have a role to play beyond the first charge. To investigate this further, we followed another particle throughout its second cycle (Fig. 4). Direct visual assessment of the electrochemical activity maps at different SOC clearly reveals not only the spatial heterogeneity but also the temporal asynchronicity. Two domains with distinct behavior are identified (D5 and D6). D5 initiates its charging process at the beginning of the cell charge and reaches to its maximum SOC well before the end of the cell charge. Interestingly, while the whole cell is still being charged between time stamps 3 and 4, D5 already reverse its trend and starts to demonstrate a local discharging behavior. D6, on the other hand, shows a considerable delay. Its charging behavior starts around time stamp 3 and persists until time stamp 6, which is already significant into the cell discharging process. The evolution of the probability distribution of the local electrochemical activity is plotted in Fig. 4b, showing that the orange peak (for D6) is lagging and following the purple peak (for D5). These asynchronous activities of D5 and D6 are schematically

illustrated in Fig. 4c. The charging or discharging of local domains is essentially the effect of lithium-ion diffusion from high-concentration to low-concentration regions. The microscopic behavior could deviate from the macroscopic observation at some local regions given the high level of chemical heterogeneity and domain asynchronicity. It shall be noted that the cell-level electrochemical signals (e.g., the cell current and voltage profiles) represent a statistically accumulated effect of all the active particles in the system. Therefore, such local discrepancies can only occur as outliers in the system, but they could be functionally very important. Collectively, these domain dynamics govern the cell behavior. A well-designed cathode material that enables a smooth interdomain lithium-ion diffusion and charge transfer and, thus, facilitates a rapid domain equilibration could effectively suppress the stress accumulation and prevent particle cracking.

## Discussion
In summary, operando optical microscopy and synchrotron-based CMCD techniques are employed for investigating local electrochemical activities at sub-secondary-particle scale. Our study reveals that the lithium-ion transport process in layered oxides cathode is not only heterogeneous but also asynchronous. In the first charge, the asynchronicity is ubiquitous within and among different active particles, likely due to the large variation of domain activation energy and local current density. Interestingly, the lithium-ion transport pattern features a near-surface point onset followed by a progressive reaction front propagation, which is distinct from the traditionally conceived

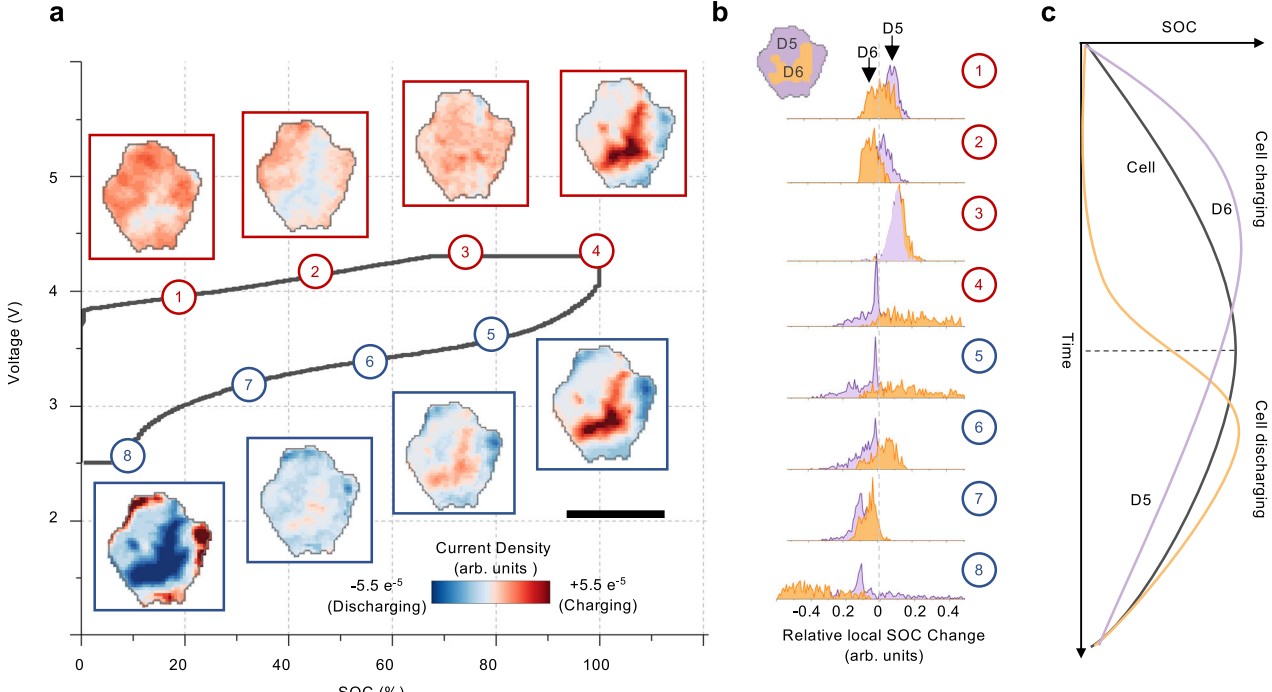

**Fig. 4 | Asynchronous domain dynamic and equilibration within an NMC particle. a** The evolution of local electrochemical activity of a selected particle over the second charge/discharge cycle. The numerical labels 1-8 represent the states at different state of charge (SOC, Red represents charging, blue represents discharging). **b** Probability distribution of local charging and discharging behavior for two identified asynchronous domains (D5 and D6). **c** Schematic illustration of the asynchronous domain dynamics and equilibration for D5 and D6. The scale bar is 5 μm.

shrinking-core reaction model. Within a single secondary particle, domains with very different local dynamics coexist, featuring an effect of inter-domain hysteresis, charge transfer, and equilibration beyond the first cycle.

An in-depth understanding of the microscopic dynamic processes in layered oxide cathodes is of fundamental significance. Our results clearly demonstrate an incoherent electrochemical reaction at the primary grain level with both spatial and temporal inconsistencies. This effect could be a significant but long-overlooked contributor to the chemomechanical disintegration of the electrodes and particles. At the same time, our finding reveals an intricate relationship between heterogeneity and asynchronicity in layered cathode for lithium-ion battery. The domain asynchronicity appears to be a very significant phenomenon in the cell activation cycle and is suppressed as the cycling continues. In contrast, heterogeneity persists and progressively evolve upon prolonged cycling. These two distinct factors are intertwined, and they affect cell performance through different mechanisms. Understanding and controlling these aspects are vital for optimizing the longevity of lithium-ion battery cathodes. This work highlighting the importance of a systematic multi-scale structural design. In particular, integrating the design of particle porosity[38,39], composition distribution[40,41], and spatially heterogeneous doping[42,43] could potentially offer an effective approach to address the detrimental effects associated with the heterogeneity and asynchronicity of domain activities.

From the technical perspective, we highlight that the herein demonstrated CMCD approach is a strong extension to the conventional XRD method. It can capture microscopic domain dynamics with sensitivity to both the lattice configuration and the domain rigid/non-rigid motions. The optical microscopy, on the other hand, is highly versatile and cost effective. It can directly reveal the evolutions of local current and reaction front propagation in real-time. These methods are crucial for studying non-equilibrium processes on the appropriate length and time scales.

## Methods

### Electrode making

NMC cathodes were prepared following the protocol described below. First, a slurry was prepared using as-received NMC powder ($LiNi_{0.5}Mn_{0.3}Co_{0.2}O_2$, Toda America), carbon black (CB, Denka), polyvinylidene fluoride (PVDF, Solvay, 5130), and n-methyl-2-pyrrolidone (NMP, Sigma Aldrich). For the optical imaging experiment, the slurry was cast onto a battery-grade aluminum sheet using a slot-die coating method, with an aerial loading of 12.5 mg per $cm^2$ and with weight ratios of 90 wt.% NMC, 5 wt.% PVDF, and 5 wt.% CB. To expose a flat and smooth surface (Fig. S3) suitable for high magnification optical imaging, a strip of 0.8 $cm^2$ area was cut out of the cathode and subjected to ion polishing using a JEOL IB-19500CP ion polisher with a rotating stage. In the CMCD experiment, the thickness of the electrode was controlled at 10 μm.

### CMCD experiment

CMCD experiments were carried out at the 34-ID-C beamline of the Advanced Photon Source (APS). In our experiment, we used 9 keV monochromatic beams. The synchrotron X-ray beam was focused using a pair of Kirkpatrick–Baez (KB) mirrors to approximately 1 × 1 $μm^2$ illuminating the NMC ($LiNi_{0.5}Mn_{0.3}Co_{0.2}O_2$) samples. The measurement was carried out using a 10 μm thick NMC electrode, situated within a coin cell setup designed for operando CMCD. This configuration involves coin cells with central through-holes of 1–2 mm diameter in both cases, sealed with polyamide tape. The cell is assembled in order inside, starting from the followed by cathode, electrolyte-soaked separator, lithium foil anode (Commercial grade, 99.9% Li, Sigma Aldrich), spacer and concluding with a spring. CMCD patterns were acquired using a Timepix photon-counting detector, mounted at -1950 mm downstream from the sample. Full rocking curves were obtained around the (003) Bragg reflection, and 2D CMCD patterns were collected at 2θ angles of 18.6° (with a resolution of Δθ = ± 0.15°). An automatic denoising and background correction method was

implemented to improve data quality. Operando observation of the sample evolution was achieved by rocking the sample in 0.0025° steps around the Bragg peak while the coin cell was cycled at a C/5 current rate (1C = 200 mA/g) with a working window of 2.5–4.5 V using an eight-channel MACCOR battery cycler during the measurement process in room temperature ~25 °C.

The current implementation of CMCD has its limitations. For example, when significant morphological change occurs, it becomes very difficult to track the diffraction spots throughout the entire experiment. The physical translation, rotation, and deformation of active particles and electrodes can be induced by a variety of chemical and physical processes during the cell operation. This challenge could potentially be addressed through the implementation of simultaneous sample rocking and scanning, for which a synchrotron source with higher brilliance and coherence is desirable.

## Operando optical experiment

The cathode sample was dried overnight in a vacuum oven at 80 °C to remove any residual moisture. It was then transferred to an argon-filled glovebox with $O_2$ and $H_2O$ concentrations below 1 ppm and fixed to the center of a fluid cell (as shown in Fig. S5) as the working electrode. A non-contacting Li metal ribbon (99.9% Li, Sigma Aldrich) was wrapped around the cathode as the counter electrode (C.E.). The cell was filled with a non-volatile electrolyte consisting of 0.75 M of $LiPF_6$ in a 1:1 weight ratio of propylene carbonate (PC, Sigma Aldrich) and ethylene carbonate (EC).

Electrochemical tests were conducted using a potentiostat (VersaSTAT4, Princeton Applied Research, Ametek). Optical images were taken every minute at fixed positions using an integrated microscope (Keysight G200 nanoindenter). The tests were performed at C/20 (1C = 200 mA/g) with a working window of 2.5–4.5 V in temperature ~25 °C. A constant potential was maintained at these cut-off voltages until the current dropped to 1/5 of the galvanostatic current to ensure that the real SOC meets our goal. Image sets process detail can be found in supplementary materials.

## Data availability

All data are available in the main text or the supplementary materials. Source data are provided with this paper.

## Code availability

The data processing method has been described in the Supplementary Information. Should any specific MATLAB code can be obtained upon request from the corresponding author.

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

## Acknowledgements
Use of the Stanford Synchrotron Radiation Lightsource, SLAC National Accelerator Laboratory, is supported by the U.S. Department of Energy, Office of Science, Office of Basic Energy Sciences under Contract No. DE-AC02-76SF00515. This research used resources of the Advanced Photon Source, US Department of Energy (DOE), Office of Science User Facility, operated for the DOE Office of Science by Argonne National Laboratory under contract no. DE-AC02-06CH11357. The work at the Central South University was sponsored by the National Natural Science Foundation of China (52172264) and Fundamental Research Funds for the Central Universities of Central South University.

## Author contributions
Conceptualization: Y.L., F.L., L.L., K.Z.; Investigation: Z.X., N.S., F.W., P.P., F.L., L.L., K.Z., Y.L.; Methodology: Z.X., Y.L.; Resources: L.L., K.Z., Y.L., F.W.; Supervision: Y.L., K.Z., L.L., F.W.; Writing—original draft: Z.X., K.Z., L.L., Y.L.; All authors reviewed, edited, and approved the manuscript.

## Competing interests
The authors declare no competing interests.
