## [Peer Review File · Nature Communications]

REVIEWER COMMENTS

Reviewer #1 (Remarks to the Author):

This work submitted by Liu et al. to the Journal of Nature Communications reported the asynchronous charge-discharge behavior of polycrystalline NMC. This paper is potentially useful but filled with problems that need to be addressed before being accepted in Nature Communications. The following are my concerns.

1. What portion of the Debye-Scherrer ring pattern was chosen to characterize this primary grain diffraction collectively? And how this resolution impacts the fidelity of the conclusion being made.
2. What is the time span between T_i and T_{i-1} ? How to trace the movement of a particular grain figure 1c-1h especially when there is electrode deformation?
3. Since operando optical microscopy was used to visualize the asynchronous domine dynamics, it would be better to elaborate on the working principle of the relationship between optical intensity and SOC. In Figure s8, it seems that a maximum of the curve at 40% SOC, which has a similar intensity as 100% SOC, how is this going to impact the interpretation of the true SOC in particles.
4. In Figure 2c, if every particle only takes 8 mins to reach 100%SOC, where does the current go for the rest 15hs when a C/15 rate is used. Or it suggests only this particular particle reaches 100% SOC in 8 mins.
5. What is the difference between Figure 2c and Figure 3a, why in Figure 2c, the particle reach 100%SOC in 8 mins while takes 15hrs for Figure 3a?
6. What is the driving force for this asynchronous charge-discharge phenomenon, and how does this phenomenon develop over cycles?
7. Another question that might be out of the scope of this study, how does this affect single crystal particles, any experiment on this or any comments?

Reviewer #2 (Remarks to the Author):

Layered oxide cathodes are quintessential components in lithium-ion batteries, underpinning their energy storage efficacy. This investigation delves into the intricate microscopic dynamic processes inherent to these cathodes. By adopting an unconventional XRD approach, Coherent Multi-Crystal Diffraction (CMCD), as well as optical microscopy techniques, the research provides interesting insights into the primary grain behaviors in operando cell. This initiative is of scientific significance, as it adeptly elucidates two critical aspects: lattice evolution and localized SOC development, both of which are useful for comprehending the underlying dynamic and equilibration of layered oxide. Their experimental observations are further bolstered by meticulous theoretical calculations, adding mechanistic insights. This work reveals a different picture of the dynamic of layer oxide cathode. This manuscript is well written, I recommend its publication in Nature Communications. But the following issues should be addressed.

1. The authors could clarify that the intensity of the optical images can reflect the localized State of Charge (SOC). Please also discuss the underlying reason of this.
2. CMCD is relatively new to the battery community. The authors are suggested to provide more information of this method. Especially, compared with traditional XRD, what is the fundamental difference and how the samples shall be prepared differently. What is the limitation of this technique for battery research? Perspectives on the future technical developments will also be useful to the community.
3. The author refers to CMCD using two different names: 'coherent multi-crystal X-ray diffraction' and 'coherent multi-crystal diffraction'. These names should be made consistent.
4. The authors have used terms like 'lithium cell', 'lithium battery', 'lithium ion battery' and 'lithium-ion battery' interchangeably throughout the paper. These terms should be made consistent.
5. In the discussion, the author mentioned that they analyzed nearly 100 particles and found that the 'shrinking-core model' is invalidated for these particles. What is the limiting factor for analyzing even more particles? The reaction heterogeneity could exist at a larger scale, i.e., across different regions of the entire electrode. Is there a way to sample an even larger area for better statistical significance?
6. The authors reported that during the initial charging process, regardless of the shape and size of the particles, they all complete charging around 10 minutes. Even in abnormal particles, the charging time for each domain is also 10 minutes. What is the scientific basis and significance behind this?
7. In the High-throughput Particle Analysis and Clustering section, the author needs to explain in detail how to isolate and cluster for each particle. What is the principle of clustering?
8. When the authors first mentioned FEM, they did not provide its full name. It would be appreciated if the authors could rectify this.
9. In the FEM discussion section, the authors frequently used the term 'heterogeneity', emphasizing its importance. The heterogeneity has been broadly observed and reported. For this paper, it will be more useful if the authors could provide an in-depth discussion on how the heterogeneity relates with asynchronicity, which is the main finding of this work.
10. The findings are interesting. Could the authors further expand on the potential implications of this research for layered oxide cathodes?

Reviewer #3 (Remarks to the Author):

The article by Xue et al describes by means of optical microscopy and coherent multi-crystal diffraction (CMCD) the finding of multiple domains reacting in NCM cathodes with local (spatial) and temporal inhomogeneity. The advanced characterization methods are well suited for such investigation. Findings are interesting, albeit not entirely new. While the authors give a quite high statistical significance to their findings, which is not so common to find, I am not sure if this represent sufficient novelty for publication in Nature Communications. Moreover, while the paper is clear and well written, I find it lacks many details and oversimplifies some aspects, which may prove problematic towards reproducibility. Perhaps the paper can be reconsidered after major revision, but at the moment I would not recommend publication. Below is a list of more detailed comments/issues.

- The NCM materials studied completely lacks any information and characterization. I believe not even the composition is mentioned. Not to talk about morphology or synthesis conditions. Morphology, in particular, would be expected to play a very important role in the findings of the paper. It would be much more useful and informative if the authors provided, among other info, SEM images and discussed their expectations when moving to polycrystalline materials to single crystalline ones (or the other way round).

- How can the authors distinguish rotations and shrinkage Fig 1 e and g?

- I also wonder about the significance of the sequential reaction observed in the CMCD experiments, with respect to real "industrial" electrodes. High loading, calendaring and achieving significant thicknesses (10 micron is a very thin electrode) are likely to significantly impact the results.

- The authors say that that: "This anisotropic delithiation pattern could potentially be attributed to the irregularity in the particle shape²⁸, non-uniform carbon-binder contact^{20,29,30}, grain structure³¹, compositional variations³² or electrode heterogeneities^{33, 34}, which are often purposely tuned to adjust the cell behavior." I think this strongly limits the learning of this paper. The anisotropy has been observed in many other papers, cited by the authors. But here there is no suggestion or insight as to what is causing the effect in the paper. And with no info on the cathode, it is hard to speculate on this.

- Why did the authors go from 4 domains observed experimentally in a particle, to 7 domains in the simulations?

- How are the domains separated? Can the authors observe if these are twins, antiphase boundaries, or which kind of other 2D extended defect? Is there any experimental observation related to this, and how is this instead implemented in the modeling?

- The authors often talk about “damage”, which is not really a scientific term. How is it defined exactly? And how is it quantified? I find the lack of details and explanations such as this quite problematic.

- I really don't understand how it is physically possible that a particle domain can discharge while the externally imposed current demands charge. This is totally counter intuitive, not to say unphysical. “D5 initiates its charging process at the beginning of the cell charge and reaches to its maximum SOC well before the end of the cell charge. Interestingly, while the whole cell is still being charged between time stamps 3 and 4, D5 already reverse its trend and starts to demonstrate a local discharging behavior” The authors should spend a lot more time on this point and try to propose a physically meaningful explanation.

- In Figure S6, in the charged state, one can clearly distinguish 2 phases crystallographically, but the authors do not discuss this at all.

- I find the modeling part rather concise. The chose model appears very Convenient, with a single domain of the particle attached to the carbon. In a real material this would not happen, ie typically the carbon in the composite (having a very large surface area) covers the cathode particles rather homogeneously. And the particles are densely in contact with each other. I find the authors should repeat the modelling with other configurations of the composite to strengthen their findings.

Reviewer #4 (Remarks to the Author):

I have carefully reviewed the manuscript titled "Asynchronous Domain Dynamics and Equilibration in Layered Oxide Battery Cathode" submitted to Nature Communication. The author utilized a diffraction method termed coherent multi-crystal X-ray diffraction (CMCD) combined with an optical microscope to probe the chemical dynamic processes in operating local domains of the layered oxide cathode at the microscale. The model system they used is the commercialized $\text{LiNi}_x\text{Mn}_y\text{Co}_z\text{O}_2$ (NMC), where $x+y+z=1$, which is well understood and has been broadly studied.

The author managed to observe the onset of lithiation process at the surface of individual particles, contrasting the conventional shrinking-core model. However, despite the systematic study and concrete evidence, the current finding is not new, and the novelty of CMCD is overestimated at its current stage. I would like to provide my comments and concerns on the manuscript.

My biggest concern is the main focus of this study. While the author highlighted previous studies to justify the necessity of introducing CMCD into studying the dynamic process of NMC during operation, I can only find one set of data collected from CMCD, which is shown in Figure 1. In fact, most of the results that provide direct information on the local dynamic behavior come from optical images and subsequent analysis presented in the rest of the manuscript. Therefore, while the author states that "From a technical perspective, we highlight that the herein demonstrated CMCD approach is a strong extension to the conventional XRD method," it is not convincing

Additionally, from my own perspective, CMCD is still an averaging X-ray diffraction-based technique as it does not provide spatial resolution and is unable to resolve local information in real space. This is why the authors further employed operando optical microscopy techniques to observe the local information of a single NMC particle. I wonder what new information exactly the CMCD technique provided if the optical microscope can already fulfill the requirements?

Most importantly, the main finding of this paper has been reported with similar results and techniques using an optical microscope. For instance, Xu et al. (2022) observed kinetically induced lithium heterogeneities in single-particle layered Ni-rich cathodes through operando visualization. I suggest that the authors provide a discussion differentiating the new findings in their manuscript.

Point-to-point revision summary for

Article NCOMMS-23-36968: *Asynchronous Domain Dynamics and Equilibration in Layered Oxide Battery Cathode*

Reviewer comments are color-coded in black

Author responses are color-coded in blue

Manuscript revisions are color-coded in red

We are thankful to all the four reviewers for their efforts in evaluating our submission and for making constructive comments, which we find very useful for our revision. We are delighted that the first and the second reviewers have suggested the acceptance of our manuscript, pending suitable revisions in response to the questions raised. We also appreciate that both the third and the fourth reviewers have independently provided positive evaluations of the technical aspects of our work, including the statistical significance.

In addition to the high-level assessments, specific questions were raised and are now addressed in detail. With these valuable inputs, we believe that the revised manuscript is significantly improved and the impact of our work is more clearly communicated. We are very grateful for this review process. Our point-to-point response is summarized below, and we look forward to further interactions with all the reviewers and the editorial office.

Reviewer #1

This work submitted by Liu et al. to the Journal of Nature Communications reported the asynchronous charge-discharge behavior of polycrystalline NMC. This paper is potentially useful but filled with problems that need to be addressed before being accepted in Nature Communications. The following are my concerns.

We thank Reviewer 1# for the high-level assessment and the specific suggestions related to our submission. The detailed responses are listed below and are incorporated in the revised manuscript.

1. What portion of the Debye-Scherrer ring pattern was chosen to characterize this primary grain diffraction collectively? And how this resolution impacts the fidelity of the conclusion being made.

CMCD is a technique that bridges the single-crystal diffractive imaging and powder XRD. In our CMCD experiment, X-rays are monochromatized using a double-crystal Si (111) monochromator, and then focused onto the sample with a Kirkpatrick-Baez mirror system. Detectors (CCD or Timepix) capture the X-rays scattered from the sample to obtain data that reveals the sample's lattice

configuration. In theory, we can sample any part of the Debye-Scherrer ring with tunable resolution and reciprocal space coverage. However, in practice, there are geometric constraints on the spatial arrangement of setup, such as the mirror-to-sample distance, the sample-to-detector distance, and the angular position of the CCD. Working with constraints at this specific beamline, prior to the operando experiment, we carried out a systematic optimization process to determine the suitable configuration. A few factors were considered in this process: 1) we aim to optimize the overall intensity of the XRD signal on the CCD detector, 2) we aim to maximize the solid angle coverage while maintaining the resolvability of individual diffraction spots, 3) we aim to optimize the mechanical stability of the setup to rule out any artifacts that might be induced by the hardware. Our experiment collected the lower portion of the [003] ring. If a different portion of the Debye-Scherrer ring is selected, the semi-statistical behavior shouldn't be influenced because it should be consistent with XRD results. Similarly, if a different location of the sample is measured, the position of the [003] ring shouldn't differ from the current results. The individual behaviors of primary particles might vary from different sampling. In general, an asynchronous behavior of individual particles was observed regardless of the sampling location.

We would like to echo the comment from Reviewer 1# on the importance of resolution in a CMCD experiment. In our experiment, we utilized industry-relevant NMC cathode particles, which are in the form of several-micron-sized secondary particles. Within the secondary particles, the primary domains are small (at ~100 nm level) and their crystallographic orientations are random. In our experiment, the X-ray spot is about 1 μ m full-width-half-maximum in the lateral direction. Therefore, only tens of primary particles are oriented in the detection geometry. Each of these primary grains will cast a diffraction spot on the recorded (003) ring and it requires a decent resolution to resolve these individual diffraction spots and to follow them as the material is being electrochemically cycled. This is exactly the reason for us to carefully achieve a balanced compromise between the conflicting desires of a large solid angle coverage and a high resolving power.

We confirmed the fidelity of our data by comparing our results against the conventional XRD results that are quite broadly reported in the literature. Our diffraction pattern of the (003) peak (Figure 1b) demonstrates a trend that is consistent with the reported behavior of similar materials (DOI: 10.1016/j.ssi.2020.115520), where the lattice undergoes expansion first and then shrink during the charging process (as shown in Figure R1). Thus, we are confident that our CMCD results are reliable and robust.

Figure R1 In situ XRD of (003) diffraction peak from in-situ NCM half-cell reported in the literature (DOI:10.1016/j.ssi.2020.115520) and powder XRD of (003) diffraction peak from CMCD measurement and the corresponding charge-discharge curve of the cell in this work (Figure 1b in the current paper).

Finally, as a forward-looking note, we add here that it is technically possible to cover a large solid angle with high resolving power by simultaneously incorporating multiple area detectors. This, however, would not only require significant experimental resources and developments, but also adds challenges in the data curation process. We envision that further technical developments and investments will be made in this at synchrotron beamlines for material characterization. It will certainly be beneficial for future works in this field using CMCD technique.

2. What is the time span between T_i and T_{i-1} ? How to trace the movement of a particular grain figure 1c-1h especially when there is electrode deformation?

The time span between T_i and T_{i-1} is 10 min. This is chosen during the process of experimental optimization. We initially ran the XRD measurements with a much higher frame rate and concluded that a high frame rate would significantly increase the dose to the sample but would not add more useful information to our experiment. Therefore, as a dose management measure, we settle to the experimental protocol described in the manuscript.

Given the purposely selected low charging rate, the 10 min time span is a reasonable temporal resolution for this experiment. As we shown in the supplementary video, most of the diffraction peaks can be visually followed. The directions of the peak motion, however, have different implications. As we illustrated in Figure R2 (refer to Figure 1g), the peak trajectory vector can be divided into two independent components: 1) being normal to the ring, changing 2θ and 2) being along the tangential direction, changing η . The first component is relevant to the lattice breathing and the second component is resulted from the physical rotation of the grain. Therefore, the real-space motion of the primary grains can be inferred by analyzing the trajectory of the diffraction peak.

Figure R2: Schematic illustration of CMCD pattern depicting lattice contraction and domain rotation in the operating NMC lithium-ion cell (Figure 1g in the current paper)

To carry out this analysis with high efficiency and accuracy, algorithms developed in the computer vision community, e.g., algorithms for feature recognition and image registration, can be leveraged. Our group has devoted significant effort into this area over the last decade. Software and algorithms, including the source code resulted from our efforts in this direction are made publicly available in our group's GitHub repository.

We clarify that, if the grain motion and, thus, the electrode deformation is large, we could lose track of the diffraction peak. This actually happened to some of the peaks in our data (see the supplementary movie S1). Fortunately, there is a sufficient number of peaks that are trackable over the region of interest. Going forward, we believe that the implementation of simultaneous sample rocking will be a useful approach to ensure that more peaks are trackable over a larger region of interest. Further algorithm developments are needed to facilitate a higher degree of automation with good accuracy.

3. Since operando optical microscopy was used to visualize the asynchronous domain dynamics, it would be better to elaborate on the working principle of the relationship between optical intensity and SOC. In Figure s8, it seems that a maximum of the curve at 40% SOC, which has a similar intensity as 100% SOC, how is this going to impact the interpretation of the true SOC in particles.

A first-order visualization of the relationship between the optical intensity and the SOC is shown in Figure R3, in which, we plot the averaged intensity for all the particles that were segmented in our data as a function of the SOC. The data in Figure R3 is from the second charge, which shows a more synchronized behavior. Therefore, we largely rule out the particle-to-particle asynchronicity in this plot and associate the optical intensity to the SOC with better fidelity. A positive correlation is clearly observed, which sets the basis of our analysis.

Figure R3 Evolution in the average particle intensity during the second charge (Figure S1d in the supplementary materials of the current paper)

The plot in Figure R3, however, needs to be normalized to account for the original image contrast that is irrelevant to the electrochemistry. Therefore, we plotted the particles' normalized intensity variation over the 2nd charge in Figure R3. Thanks to this comment from Reviewer 1#, we realize that this normalization process exaggerates the data points with high noise level, which could lead to a misconception of the overall trend, clearly contradicting what we observe in the raw data in Figure R3. Therefore, here we present the same plot excluding particles with insufficient signal to noise ratio.

Figure S8. Linear regression plot illustrating the relationship between particles' optical density and the SOC during the second charge. Each color represents a unique particle. The linear regression reveals a positive correlation with a slope of 2.66 and 0.75 with R-squared value of 0.86 and 0.59 respectively.

It is interesting to observe that, although there is an overall monotonic increase of optical intensity as the SOC increases, a difference in the slope occurs above and below around SOC of 30%. Currently, the cause for different slopes is not clear and is beyond the scope of this study. We believe that additional experiments to

analyze a larger number of particles together will provide a more accurate relation between optical reflective intensity.

4. In Figure 2c, if every particle only takes 8 mins to reach 100%SOC, where does the current go for the rest 15hs when a C/15 rate is used. Or it suggests only this particular particle reaches 100% SOC in 8 mins.

This is indeed the asynchronous behavior of the particles and domains that we are reporting. Based on our observation of over a hundred particles, each of them completes its own charging process within around 10 minutes. This is a ubiquitous effect in the initial charging (cell activation) process. All the particles collectively contribute to the cell current in a sequential manner during the first charge. The cell charging was at a C/15 rate, but locally, the charging rate during the cell activation process can be much higher.

5. What is the difference between Figure 2c and Figure 3a, why in Figure 2c, the particle reach 100%SOC in 8 mins while takes 15hrs for Figure 3a?

According to our analysis of nearly 100 particles, almost all of the particles behave similarly and we selected a representative particle for analysis in Figure 2. However, through the clustering analysis of the particles, a process depicted in Figure R4, we identify a few particles as outliers with behaviors that are different from the majority. Therefore, we highlighted one of them in Figure 3. Figure 3a shows that this particle is not only charged by multiple local current pulses, but also has one local domain charged very slowly over the course of 15 hours. We believe that this is a useful observation, which is worth reporting and warrants follow-up efforts.

Figure R4 Workflow for analyzing and classification of the individual particle's SOC evolution. (Figure S9 in the supplementary materials of the current paper)

6. What is the driving force for this asynchronous charge-discharge phenomenon, and how does this phenomenon develop over cycles?

The driving force for this asynchronous domain dynamics can be attributed to the heterogeneity in local kinetic reaction conditions and activation energy barriers. In a real-world electrode, this can be affected by inhomogeneous contact with the carbon binder domain, local differences in composition, and cracks, among others. This asynchronous dynamic behavior will eventually trend towards synchronicity, as shown in Figure R5 and predicted in our previous publication (Figure R6, DOI: 10.1126/science.abm8962). The existence of the reported sequential

particle/domain charging process during the cell activation process, however, could critically affect the cell longevity and is not well understood.

Figure R5 Evolution in the averaged particle intensity during the first (a) and second charge (b). (Figure S1a and d in the supplementary materials of the current paper)

Figure R6 Finite element analysis of the electrochemical activity and mechanical damage in the NMC cathode. (A) Illustration of the composite model during the charging process in the battery. (B) Normalized Li concentration profiles depict the inherent heterogeneity of the system during the first charging process with respect to the normalized time t/τ , where t is the real time in Li reactions and $\tau = 720$ s is the theoretical time to reach the full capacity of NMC. Although the particles start with the same state of charge, Li concentration differs at the end of the first charge process. (C) The variation of Li concentration profiles among three NMC particles. The overall trend demonstrates the tendency toward a synchronized behavior. (D) The damage profiles for three NMC active particles diverge near the end of the first charge process. With the progression of the cycling process, the damage profiles for all three particles converge. (E) Each particle's deviation from the mean damage profile (the black dashed line). This figure is from our previous publication (DOI: 10.1126/science.abm8962).

7. Another question that might be out of the scope of this study, how does this affect single crystal particles, any experiment on this or any comments?

We appreciate this forward-looking comment on the single-crystal particles, which is relevant yet different. As it has been broadly reported, for systems with polycrystalline particles, there is a significant chemical heterogeneity at both the electrode level and the particle scale. At the electrode level, the micromorphology

plays a dominating role (DOI: 10.1038/s41467-020-16233-5, as shown in Figure R7). Within an individual particle, the grain boundaries and cryptographic orientations have a more direct impact (DOI: 10.1038/s41467-019-13884-x, as shown in Figure R8).

Figure R7 Selected of slices of one NMC particle rendered from high-resolution hard X-ray nano-tomography data. The images reveal an inhomogeneous contact between the active particle and the carbon binder domain. The scale bar in is 10 μ m. This figure is reused from our previous publication (Nature Communications 11, 2310 (2020))

Figure R8 a 3D Ni valence state rendering from TXM data of a cycled polycrystalline NMC particle, b representative region of the 3D Ni valence state distribution, and c 2D nanodomain valence gradient of it. This figure is reused from our previous publication (DOI: 10.1038/s41467-019-13884-x)

For single-crystalline particles, on the other hand, the particle's internal structure is simplified. Therefore, we envision that the activation of each single-crystal particle during the first charge will be even more heavily governed by the local electrode morphology. We have not yet conducted any experiment on this using the approach described in this paper. We believe that this is a topic of interest for follow-ups.

Reviewer #2

Layered oxide cathodes are quintessential components in lithium-ion batteries, underpinning their energy storage efficacy. This investigation delves into the intricate microscopic dynamic processes inherent to these cathodes. By adopting an unconventional XRD approach, Coherent Multi-Crystal Diffraction (CMCD), as well as optical microscopy techniques, the research provides interesting insights into the primary grain behaviors in operando cell. This initiative is of scientific significance, as it adeptly elucidates two critical aspects: lattice evolution and localized SOC development, both of which are useful for comprehending the

underlying the dynamic and equilibration of layered oxide. Their experimental observations are further bolstered by meticulous theoretical calculations, adding mechanistic insights. This work reveals a different picture of the dynamic of layer oxide cathode. This manuscript is well written, I recommend its publication in Nature Communications. But the following issues should be addressed.

We thank reviewer #2 for his/her positive assessments of our work. We find the specific comments listed below very useful and constructive.

1. The authors could clarify that the intensity of the optical images can reflect the localized State of Charge (SOC). Please also discuss the underlying reason of this.

A first-order analysis of the relationship between the optical intensity and the SOC is shown in Figure R9, in which, we plot the averaged intensity for all the particles that were segmented in our data as a function of the SOC. The data in Figure R9 is from the second charge, which shows a synchronized behavior. Therefore, we largely rule out the particle-to-particle asynchronicity in this plot and associate the optical intensity to the SOC with better fidelity. A positive correlation is clearly observed, which sets the basis of our analysis. With a careful normalization process (new Figure S8, please also refer to Question 3 from Reviewer 1#), it is interesting to observe that, although there is an overall monotonic increase of optical intensity as the SOC increases, a difference in the slope occurs above and below around SOC of 30%.

Figure R9 Evolution in the average particle intensity during the second charge (Figure S1d in the supplementary materials of the current paper)

The mechanism for such a difference is unclear at this writing. Nevertheless, the overarching dependence of the optical reflective intensity on SOC is because of the evolving dielectric property of NMC upon charging. In the writing of the dielectric function

$$\varepsilon(\omega) = \varepsilon_1(\omega) + i\varepsilon_2(\omega)$$

ε_1 represents the ability of electrons to release photons from the excited state to the lower energy state, and ε_2 represents the light absorption process of electrons

from the valence to the conduction bands. Based on the dielectric function, the optical reflectivity $R(\omega)$ can be determined by

$$R(\omega) = \frac{[n(\omega) - 1]^2 + k(\omega)^2}{[n(\omega) + 1]^2 + k(\omega)^2}$$

where $n(\omega)$ and $k(\omega)$ are the reflection index and extinction coefficient, respectively, which are calculated from ϵ_1 and ϵ_2 . We are conducting preliminary density function theory (DFT) based modeling to understand the optical property of NMC cathode. The Figure R10 shows the reflectivity of NMC532 during extraction, where delithiation induces a red shift of reflectivity in the a/b directions but purple shift in the c direction.

Figure R10 DFT analysis of the dependence of reflectivity as a function of wavelengths of light for NMC 523 with varying Li content. (Unpublished)

The following Figure R11 summarizes the peak intensity as a function of SOC, in which reflectivity shows a linear increase during the charging process, which is consistent with the experimental observation.

Figure R11 The dependence of the peak intensity as a function of lithium content in different components of NMC materials. (Unpublished)

- CMCD is relatively new to the battery community. The authors are suggested to provide more information of this method. Especially, compared with traditional XRD, what is the fundamental difference and how the samples shall be prepared differently. What is the limitation of this technique for battery research? Perspectives on the future technical developments will also be useful to the community.

Coherent Multi-Crystal Diffraction (CMCD) is indeed a relatively new technique. It leverages coherent X-ray beams to study the structural dynamics of battery materials, offering a more detailed insight into the lattice dynamics with individual grain resolving power compared to traditional XRD methods. Some of our coauthors have been actively developing the CMCD method and a method paper describing the applications of coherent X-ray diffraction techniques on battery materials has been referenced (Li J. Synch. Radiation 26, 220(2019), ref#29 in our manuscript). The fundamental difference between CMCD and traditional powder XRD technology is in the approach to collect and analyze diffraction patterns. Traditional XRD averages the diffraction signals over an area of several square millimeters on the electrode. It provides a bulk averaged information. CMCD, on the other hand, focuses to a smaller spot ($\sim 1 \mu\text{m}^2$ in this work), allowing for the study of a relatively small numbers of grains, thereby providing more detailed information on the domain dynamics.

The sample preparation is similar to XRD methods. Coin Cell, Pouch cell, AMPIX cell or other in-situ battery cells can be used for CMCD measurements. In this work, a monolayer electrode of active NMC particles were loaded into coin cells with through-holes in the shells to ensure that all primary particles participating in diffraction come from a single secondary particle.

The current implementation of CMCD has its limitations. For example, when significant morphological change occurs, it becomes very difficult to track the diffraction spots throughout the entire experiment. The physical translation, rotation, and deformation of active particles as well as electrodes can be induced by a variety of chemical and physical processes during the cell operation. This is highly relevant to the battery degradation and failure analysis. Another limitation in the current configuration is that the detector can only measure one Bragg peak at a time. This challenge could potentially be addressed using an X-ray source with a higher level of coherence and a large area detector or detector array. The implementation of simultaneous sample rocking and scanning is another viable approach to facilitate the tracking of diffraction spots with an improved robustness against the sample deformation.

We have included some of these discussions in the revised manuscript.

The current implementation of CMCD has its limitations. For example, when significant morphological change occurs, it becomes very difficult to track the diffraction spots throughout the entire experiment. The physical translation, rotation, and deformation of active particles and electrodes can be induced by a variety of chemical and physical processes during the cell operation. This challenge could potentially be addressed through the implementation of simultaneous sample rocking and scanning, for which a synchrotron source with higher brilliance and coherence is desirable.

3. The author refers to CMCD using two different names: 'coherent multi-crystal X-ray diffraction' and 'coherent multi-crystal diffraction'. These names should be made consistent.

We have made corrections accordingly and use the term 'Coherent Multi-Crystal Diffraction (CMCD)' throughout the manuscript.

4. The authors have used terms like 'lithium cell', 'lithium battery', 'lithium ion battery' and 'lithium-ion battery' interchangeably throughout the paper. These terms should be made consistent.

We have made corrections accordingly and use the term 'lithium-ion battery' throughout the manuscript.

5. In the discussion, the author mentioned that they analyzed nearly 100 particles and found that the 'shrinking-core model' is invalidated for these particles. What is the limiting factor for analyzing even more particles? The reaction heterogeneity could exist at a larger scale, i.e., across different regions of the entire electrode. Is there a way to sample an even larger area for better statistical significance?

The limitations that prevent us from analyzing even more particles using optical microscopy include 1) under the desired spatial resolution, there is a limited size of the field of view and 2) utilizing the visible light as a probe for the optically opaque electrode, we can only study the exposed electrode cross section, lacking information from the buried particles. This is one of the main reasons for our exploration and adoption of CMCD technique with X-rays. CMCD offers penetrative insights, demonstrating asynchronous behaviors of the buried primary crystalline grains. Looking forward, the correlative integration of X-ray spectro-microscopy and optical microscopy can be a viable approach for gaining further microscopic insights.

6. The authors reported that during the initial charging process, regardless of the shape and size of the particles, they all complete charging around 10 minutes. Even in abnormal particles, the charging time for each domain is also 10 minutes. What is the scientific basis and significance behind this?

Based on our observations during the cell activation process, almost all the particles' de-lithiation reactions occur asynchronously. Each of them completes its first charge within approximately ten minutes regardless of the size and shape. This asynchronous particle charging behavior can be attributed to different local characteristics associated with different particles, such as contact with the carbon binder domain and compositional variation. Interestingly, the charging time for each particle is similar. This finding is corroborated by other studies. For instance, a recent study (Energy Environ. Sci., 2023, 16, 3847-3859) using a single-particle battery configuration reported that all particles complete their respective charging process within a similar time frame, suggesting that the ion diffusion model may not be sufficient to analyze the secondary layered oxide particles (as shown in Figure R12). Moreover, another recent research (DOI: 10.1038/s41565-023-

01466-4) shows that the kinetics of (de)lithiation are controlled by competition at the surface-intermediate, unrelated to particle size. This insight could potentially aid in revising strategies to optimize battery performance, particularly during the cell activation process.

Figure R12 The dependence of the diffusion time τ_D as a function of particle diameter. This figure is reused from (Energy Environ. Sci., 2023, 16, 3847-3859)

7. In the High-throughput Particle Analysis and Clustering section, the author needs to explain in detail how to isolate and cluster for each particle. What is the principle of clustering?

In the "High-throughput Particle Analysis and Clustering" section, we delineate a method to individually isolate and cluster particles for analysis. We start with the registered image stack and labeled segmentation results to extract and scrutinize the particles one at a time. To group the particles with similar behavior, we first extract the averaged chemical state, intraparticle heterogeneity, and dynamic intensity changes of each particle. We evaluate the averaged characteristics for all the particles and then identify the ones with most distinct differences in different aspects. This process is done in an iterative manner and abnormal particles are singled out for further analysis. This approach facilitates the screening and clustering of particles, pinpointing those warranting detailed study based on their electrochemical behavior and other attributes.

We have included some of these discussions in the revised manuscript.

8. When the authors first mentioned FEM, they did not provide its full name. It would be appreciated if the authors could rectify this.

We have revised this by specifying the full term, "Finite Element Modeling (FEM)".

9. In the FEM discussion section, the authors frequently used the term 'heterogeneity', emphasizing its importance. The heterogeneity has been broadly observed and reported. For this paper, it will be more useful if the authors could provide an in-depth discussion on how the heterogeneity relates with asynchronicity, which is the main finding of this work.

"Heterogeneity" and "Asynchronicity" are interrelated characteristics in electrochemical processes. "Heterogeneity" refers to the variation in physical or chemical properties across different regions/domains of the cathode material. This phenomenon may originate from the battery manufacturing process or be due to uneven distribution of lithium-ion and the accumulation of stress during cycling, both of which can adversely impact the battery's performance and service life. The term "asynchronicity" describes the temporal disparities in electrochemical reactions or processes, where lithium-ion intercalation and deintercalation in different regions do not occur at same time, fostering asynchronous electrochemical reactions. The asynchronicity can escalate issues brought about by heterogeneity, inducing faster degradation and lower efficiency. Based on our experimental observation, the domain asynchronicity is very significant in the cell activation cycle and is suppressed over time. In contrast, the heterogeneity persists over long time and could evolve slowly upon prolonged cycling. Understanding and controlling these aspects are vital in optimizing the performance of lithium-ion battery cathodes.

We have included some of these discussions in the revised manuscript (see the red text in our response to the next question).

10. The findings are interesting. Could the authors further expand on the potential implications of this research for layered oxide cathodes?

We have further elaborated on the potential implications of our findings for layered oxide cathodes in the conclusion section of the manuscript:

"This effect could be a significant but long-overlooked contributor to the chemomechanical disintegration of the electrodes and particles. At the same time, our finding reveals an intricate relationship between heterogeneity and asynchronicity in layered cathode for lithium-ion battery. The domain asynchronicity appears to be a very significant phenomenon in the cell activation cycle and is suppressed as the cycling continues. In contrast, heterogeneity persists and progressively evolve upon prolonged cycling. These two distinct factors are intertwined, and they affect cell performance through different mechanisms. Understanding and controlling these aspects are vital for optimizing the longevity of lithium-ion battery cathodes. This work highlights the importance of a systematic multi-scale structural design. In particular, integrating the design of particle porosity^{38,39}, composition distribution^{40,41}, and spatially heterogeneous doping^{42,43} could potentially offer an effective approach to address the detrimental effects associated with the heterogeneity and asynchronicity of domain activities. "

Reviewer #3

The article by Xue et al describes by means of optical microscopy and coherent multi-crystal diffraction (CMCD) the finding of multiple domains reacting in NCM cathodes with local (spatial) and temporal inhomogeneity. The advanced characterization

methods are well suited for such investigation. Findings are interesting, albeit not entirely new. While the authors give a quite high statistical significance to their findings, which is not so common to find, I am not sure if this represent sufficient novelty for publication in Nature Communications. Moreover, while the paper is clear and well written, I find it lacks many details and oversimplifies some aspects, which may prove problematic towards reproducibility. Perhaps the paper can be reconsidered after major revision, but at the moment I would not recommend publication. Below is a list of more detailed comments/issues.

We are thankful to Reviewer 3# for the constructive feedback and for recognizing the statistical significance of our results. We address the specific questions and concerns below with our best efforts.

- The NCM materials studied completely lacks any information and characterization. I believe not even the composition is mentioned. Not to talk about morphology or synthesis conditions. Morphology, in particular, would be expected to play a very important role in the findings of the paper. It would be much more useful and informative if the authors provided, among other info, SEM images and discussed their expectations when moving to polycrystalline materials to single crystalline ones (or the other way round).

We clarified that we utilized commercial NMC532 secondary spherical particles for our research. To offer a detailed view of the morphology, SEM images have been included and can be referred to in Figure S3(b). For additional details on the morphology, we would like to refer to our previous study (Nano Letters 22, 14, 5883–5890, 2022, as shown in Figure R13), where more baseline characterizations of the material are reported.

Figure R13 Scanning electron microscopy (SEM) images of the NMC cathode (a) before and (b), (c) after surface polishing. Scale bars correspond to 10 μm . This figure is reused from our previous publication (Figure from Nano Letters 22, 14, 5883–5890, 2022)

The investigation of single-crystalline particles is beyond the scope of this work. However, this is a valuable, forward-looking comment that warrants follow-up efforts. Here we provide our perspectives on this topic.

As it has been broadly reported, for systems with polycrystalline particles, there is a significant chemical heterogeneity at both the electrode level and the particle scale. At the electrode level, the micromorphology plays a dominating role (DOI: 10.1038/s41467-020-16233-5). As it shown in Figure R14, there are significant ununiform detachments between NMC particle and carbon and binder domains (CBD) which could lead to heterogenous lithium-ion diffusion and electron transport. Within an individual particle, the grain boundaries and cryptographic orientations have a more direct impact (DOI: 10.1038/s41467-019-13884-x, as shown in Figure R15). For single-crystal particles, on the other hand, the particle's internal structure is simplified. Therefore, we envision that the activation of each single-crystal particle during the first charge will be even more strongly governed by the electrode's local morphology.

We have not yet conducted any experiment on the single-crystal systems using the approach described in this paper. We believe that this is a topic of interest for follow-ups.

Figure R14 Selected of slices from one NMC particle rendered from high-resolution hard X-ray nano-tomography data. The scale bar in is $10\mu\text{m}$. This figure is reused from our previous publication (Nature Communications 11, 2310 (2020))

Figure R15 a 3D Ni valence state distribution rendering from TXM data of one cycled NMC particle, b representative region of the 3D Ni valence state distribution, and c 2D nanodomain valence gradient of a cycled NMC particle. This figure is reused from our previous publication (DOI: 10.1038/s41467-019-13884-x)

- How can the authors distinguish rotations and shrinkage Fig 1 e and g?

As we shown in the supplementary video, most of the diffraction peaks can be visually followed. The directions of the peak motion, however, have different implications. As illustrated in Figure R16, the peak trajectory vector can be divided into two independent components: 1) being normal to the ring, changing 2θ and 2) being along the tangential direction, changing η . The first component is relevant to the lattice breathing and the second component is resulted from the physical rotation of the grain.

Therefore, the real-space motion of the primary grains can be inferred by analyzing the trajectory of the diffraction peak.

Figure R16: Schematic illustration of CMCD pattern depicting lattice shrinkage and domain rotation in the operating NMC lithium-ion cell (Figure 1g in the current paper)

For example, in Figure R17, the peak moves in the tangential direction, which indicates that there is no change in this grain's lattice parameter. The peak motion is purely induced by this grain's physical movement.

Figure R17: Schematic illustration of CMCD pattern depicting domain rotation in the operating NMC lithium-ion cell (Figure 1e in the current paper)

We would also point out that the current implementation of CMCD has its limitations. For example, when significant morphological change occurs, it becomes very difficult to track the diffraction spots throughout the entire experiment. The physical translation, rotation, and deformation of active particles as well as electrodes can be induced by a variety of chemical and physical processes during cell operation. This is highly relevant to battery degradation and failure analysis. This challenge could potentially be addressed using an X-ray source with a higher level of coherence. The implementation of simultaneous sample rocking and scanning is another viable approach to facilitate the tracking of diffraction spots with an improved robustness against sample deformation.

We have included some of these discussions in the revised manuscript.

The current implementation of CMCD has its limitations. For example, when significant morphological change occurs, it becomes very difficult to track the diffraction spots throughout the entire experiment. The physical translation, rotation, and deformation of active particles and electrodes can be induced by a variety of chemical and physical processes during cell operation. This challenge could potentially be addressed through the implementation of simultaneous sample rocking and scanning, for which a synchrotron source with higher brilliance and coherence is desirable.

- I also wonder about the significance of the sequential reaction observed in the CMCD experiments, with respect to real “industrial” electrodes. High loading, calendaring and achieving significant thicknesses (10 micron is a very thin electrode) are likely to significantly impact the results.

In this experiment, we intentionally chose a thin electrode with a single-layer of NMC particle. This configuration is selected to simplify the experiment as we would like to illuminate a small number of particles with the X-ray spot. This is very different from the conventional XRD, in which the beam footprint can be very large. We acknowledge that further complications could be induced in real industrial batteries with large thickness. A better focusing capability is needed to conduct this experiment on a thick electrode. The on-going synchrotron upgrade at the Advanced Photon Source is set to significantly increase the transversal coherence of the X-rays, which will facilitate a tighter focus with high brilliance. CMCD is expected to benefit significantly from this development, which will open vast opportunities including the investigation of industry-relevant thick electrodes.

- The authors say that that: “This anisotropic de-lithiation pattern could potentially be attributed to the irregularity in the particle shape²⁸, non-uniform carbon-binder contact^{20,29,30}, grain structure³¹, compositional variations³² or electrode heterogeneities^{33, 34}, which are often purposely tuned to adjust the cell behavior.” I think this strongly limits the learning of this paper. The anisotropy has been observed in many other papers, cited by the authors. But here there is no suggestion or insight as to what is causing the effect in the paper. And with no info on the cathode, it is hard to speculate on this.

The anisotropic de-lithiation pattern is indeed a complex phenomenon, and we acknowledge that some aspects of it have been reported in previous research (such as *Nat Commun* 7, 12372 (2016), as shown in Figure R18). In this paper, we discussed several possible factors that could contribute to the anisotropy effect, such as irregular particle shape, non-uniform carbon-binder contact, grain structure, compositional variations, and electrode heterogeneities, all of which can affect the cell performance. To truly pinpoint the driving forces for the anisotropic de-lithiation pattern, we believe that a set of systematic follow-up experiments need to be conducted. This is because batteries are intricate systems, and their behavior cannot be exclusively attributed to a single factor alone. For example, model systems of intentionally tuned electrode formation with varying packing density, particle size distribution, and compositional inhomogeneity can be compared. High-throughput experimentation and statistical analysis need to be integrated. The anisotropic dilithiation pattern is associated with the chemical heterogeneity and its interrelationship with the asynchronicity is what we would like to highlight in the current paper.

Figure R18 (a) 3D phase distribution evolution from an *operando* LiFePO₄ cell (b,c) 2D projection XANES maps obtained from two different angles. This figure is reused from (Nat Commun 7, 12372 (2016)).

As for the information about the NMC cathode material used in this study, we refer to our previous publication with various details elaborated, including the composition, electrode components, SEM images and more (Nano Letters 22, 14, 5883–5890, 2022, as shown in Figure R19)

Figure R19 Scanning electron microscopy (SEM) images of the NMC cathode (a) before and (b), (c) after surface polishing. Scale bars correspond to 10 μm. This figure is reused from our previous publication (Nano Letters 22, 14, 5883–5890, 2022)

- Why did the authors go from 4 domains observed experimentally in a particle, to 7 domains in the simulations?

There can be any number of domains inside the NMC particles, and the particle in Figure 3 contains 4. In the simulation, we tried to capture the effect of partial carbon binder coverage in different domains in the NMC particle. The higher number of domains helps us capture the similarities and dissimilarities of electrochemical behavior among domains during the charging process. Thus, seven domains were used in the simulations. We clarify here that the goal of our modeling effort is not to

replicate all the microscopic details of our experimental observation, but rather to provide a mechanistic insight.

- How are the domains separated? Can the authors observe if these are twins, antiphase boundaries, or which kind of other 2D extended defect? Is there any experimental observation related to this, and how is this instead implemented in the modeling?

We specify domains as regions of similar mechanical and transport properties inside NCM particles. Such domains can be separated by any form of defects or through physical spatial detachment. Additional characterization can be performed in follow-up studies to identify such domain boundaries' chemical and structural information. Nevertheless, from a mechanistic point of view, we aim to highlight the presence of domain boundaries and mechanical deformation during charging. In the model, the domains are attached to each other using a spring (Figure R20b). As the spring deformation increases and reaches the maximum specified value, the transport of Li through that boundary ceases.

Figure R20 (a) Schematic of a half-cell geometry used in our finite element model. The current collector at the bottom connects the external circuit to the electrode's carbon binder (black color). An NMC particle (light grey color) is divided into domains and is surrounded by liquid electrolyte (light blue color). (b) Mechanics equations to compute strain and stress in domain bulk and at the boundaries. As the spring extension reaches δ_0 the damage, Dmg begins to grow from the initial value of zero linearly with further deformation. Once the Dmg reaches the value of δ_d the boundary is completely damaged, and Li transport across the domain boundary is reduced to zero. (Figure S12 in the supplementary materials of the current paper)

- The authors often talk about “damage”, which is not really a scientific term. How is it defined exactly? And how is it quantified? I find the lack of details and explanations such as this quite problematic.

Damage is a scalar quantity that characterizes the degree of intergranular fracture of the domain boundaries within an NMC particle. The damage function is defined in

Figure R20b. It starts from a value of zero, which defines zero damage of that boundary, and as the deformation occurs at the domain boundary, the damage increases monotonically with deformation. Once it reaches the value of 1, that domain boundary is considered fully damaged. Damage controls Li transport across the domain boundaries. Charging of NMC induces mechanical deformation, leading to damage and thus affecting intra-particle Li transport.

- I really don't understand how it is physically possible that a particle domain can discharge while the externally imposed current demands charge. This is totally counter intuitive, not to say unphysical. "D5 initiates its charging process at the beginning of the cell charge and reaches to its maximum SOC well before the end of the cell charge. Interestingly, while the whole cell is still being charged between time stamps 3 and 4, D5 already reverse its trend and starts to demonstrate a local discharging behavior. The authors should spend a lot more time on this point and try to propose a physically meaningful explanation.

The phenomenon of a particle domain being discharged while the overall cell is being charged is indeed intriguing and may seem counterintuitive at first glance. However, we can elaborate on this point to provide a meaningful explanation. The charging or discharging of a local domain within the cell can be understood as the diffusion of lithium ions from regions with higher lithium-ion concentration to regions with lower lithium-ion concentration. This microscopic behavior could deviate from the macroscopic observation at some local regions.

For example, even though the entire cell is in being discharged, the local state of charge over a particular domain could be significantly higher or lower than the cell SOC due to the interplay of the asynchronicity and heterogeneity (as we observed and elaborated in this work). Therefore, lithium-ion can still diffuse from low SOC domains to high SOC domains, effectively causing the local charging behavior. The cell-level electrochemical signals (e.g., the cell current and voltage) are an accumulated effect of all the active particles in the system. Therefore, discrepancies can still occur as outliers in the system.

Similar phenomenon has also been observed in other electrochemical systems. For example, in *Nat. Nanotechnol.* 18, 790–797 (2023) (as shown in Figure R21), non-monotonic local dissolution/deposition behavior was observed in an aqueous battery system, which is another good example of the interplay between asynchronicity and heterogeneity in chemical systems.

Figure R21 The relative change in Mn concentration was determined by analyzing consecutive X-ray Fluorescence Microscopy XFM scans of an operando Lithium Manganese Oxide electrode subjected to cycling in a 2 M LiTFSI aqueous electrolyte. These scans were conducted within a voltage window spanning from -0.1 to 1.5 V versus Ag/AgCl, with a scan rate of 5 mV s⁻¹, revealing non-monotonic fluctuations in the local Mn concentration. This figure is reused from (Nat. Nanotechnol. 18, 790–797 (2023)).

We have included additional discussions on this point:

The charging or discharging of local domains is essentially the effect of lithium-ion diffusion from high-concentration to low-concentration regions. The microscopic behavior could deviate from the macroscopic observation at some local regions given the high level of chemical heterogeneity and domain asynchronicity. It shall be noted that the cell-level electrochemical signals (e.g., the cell current and voltage profiles) represent a statistically accumulated effect of all the active particles in the system. Therefore, such local discrepancies can only occur as outliers in the system, but they could be functionally very important.

- In Figure S6, in the charged state, one can clearly distinguish 2 phases crystallographically, but the authors do not discuss this at all.

The observed phase separation in Figure S6 at the charged state is a characteristic pattern in the charged state of NMC cathode, as has been reported in previous studies (Nat. Mater. 20, 991–999 2021, as shown in Figure R22; Electrochemistry Communications, 2006, 8(8):1257-1262, as shown in Figure R23). As a cathode designed to be cycled under conditions that avoid phase transitions, such phase separation is thermodynamically forbidden. It has been suggested in the literature (Nat. Mater. 20, 991–999, 2021, Figure R22) that this fictitious phase separation is a result of the heterogeneity during the charging process. This interpretation very nicely corroborates with the asynchronous reaction reported in our current paper.

Figure R22 SEM image of agglomerate particles (a) used for in-situ experiments. (b) Diffraction and electrochemistry data are combined to share the capacity axes. The intensity image plots of the (003) peak show bifurcation during fast delithiation, whereas continuous linear shifts are seen in other conditions. The line plots of the (003) peak at selected average lithium fractions show a non-unimodal evolution in the fast-delithiation condition. This figure is reused from (Nat. Mater. 20, 991–999 2021).

Figure R23 In situ XRD patterns in the (003) to (105) region of a $\text{Li}_{0.27}\text{Ni}_{0.8}\text{Co}_{0.15}\text{Al}_{0.05}\text{O}_2$ cathode during the first charge at a C/5 rate. The 2θ angles

have been converted to those corresponding to the Cu K α radiation ($\lambda = 1.54 \text{ \AA}$). This figure is reused from (Electrochemistry Communications, 2006, 8(8):1257-1262).

We have included additional discussions on this point:

To ensure that the electrode can be adjusted to the desired SOC, we conducted ex-situ XRD measurements on electrodes harvested at various charging states (see Figure S6). The results confirm that the expected electrochemical reactions did occur in our cells. Notably, at high cell SOC, a significant and thermodynamically forbidden phenomenon was observed: phase separation, as indicated by a pronounced shoulder at the (003) XRD peak. This pseudo phase separation was attributed to autocatalytic behaviors between different particles in the electrode. Subsequently, after the discharge, the peak splitting phenomenon disappears, with the (003) peak returning to its original position and shape.

-I find the modeling part rather concise. The chose model appears very Convenient, with a single domain of the particle attached to the carbon. In a real material this would not happen, ie typically the carbon in the composite (having a very large surface area) covers the cathode particles rather homogeneously. And the particles are densely in contact with each other. I find the authors should repeat the modelling with other configurations of the composite to strengthen their findings.

We would like to point out that, while the surface area of carbon and binder domains (CBD) in the composite electrode is large because of its porous structure. The distribution of CBD and its contact with active particles is not homogeneous. This non-uniformity has been widely reported in literature (Nat Commun. 11, 2310, 2020 as shown in Figure R24; Journal of Power Sources 345, 97-107 (2017) as shown in Figure R25).

Figure R24 a Selected y–z slices through an NMC particle rendered from high-resolution hard X-ray nano-tomography data. b The 3D rendering of the segmentation result of one region of interest, with the CBD set to be transparent for a better visualization of the NMC particle (orange) and the voids (gray–blue). c

The rendering of the calculated distributions of relative local electrical resistance over the surface of the particles in (b). Our modeling result suggests a strong correlation between the degree of CBD attachment and the level of calculated electrical resistance heterogeneity. d The same particle of the (c) presented without the void phase. The scale bars in (a) and (c) are 10 μm . This figure is reused from our previous publication (Nature Communications 11, 2310 (2020))

Figure R25 Carbon map (a) and superposition of the carbon and the fluorine map (b) for a detail of a partly dry film electrode. This figure is reused from (Journal of Power Sources 345, 97-107 (2017))

As shown in Figure R24 and R25, the contact between the particle and the CBD varies significantly, leading to differences in local conductivities for both electrons and lithium ions. The purpose of our modeling effort is not to replicate all the microscopic structural complexities, but to formulate a mechanistic understanding of the electrochemical consequences of inhomogeneous particle-to-CBD contact. Moreover, in this work, we focus on the primary domain dynamics and purposely construct a model with one secondary particle. The particle-to-particle interaction at the cluster/electrode level is indeed an important topic and we refer to our previous study on this topic (DOI: 10.1126/science.abm8962 as shown in Figure R26).

Figure R26 Schematic illustration of the cathode particle network evolution upon battery cycling. The electrochemical activity and degradation of all single particles co-evolve, causing a transition from particle activation to electrochemical segregation to electrode synchronization. Colors represent different activation states, and the same color indicates that particles are coevolving. This figure is reused from our pervious publication (DOI: 10.1126/science.abm8962).

I have carefully reviewed the manuscript titled "Asynchronous Domain Dynamics and Equilibration in Layered Oxide Battery Cathode" submitted to Nature Communication. The author utilized a diffraction method termed coherent multi-crystal X-ray diffraction (CMCD) combined with an optical microscope to probe the chemical dynamic processes in operating local domains of the layered oxide cathode at the microscale. The model system they used is the commercialized $\text{LiNi}_x\text{Mn}_y\text{Co}_z\text{O}_2$ (NMC), where $x+y+z=1$, which is well understood and has been broadly studied. The author managed to observe the onset of lithiation process at the surface of individual particles, contrasting the conventional shrinking-core model. However, despite the systematic study and concrete evidence, the current finding is not new, and the novelty of CMCD is overestimated at its current stage. I would like to provide my comments and concerns on the manuscript.

We are thankful for the overview of our work by Reviewer 4#.

However, we respectfully disagree with the comment that our findings are not new. Specifically, we would like to highlight our observation and understanding of the interplay between chemical heterogeneity and temporal asynchronicity. The domain asynchronicity appears to be very significant in the cell activation cycle and is suppressed over time. In contrast, the heterogeneity persists over long time and progressively evolve upon prolonged cycling. To our knowledge, this relationship is not well understood and warrants future follow-up efforts.

The specific comments listed below are valuable and are addressed with our best efforts. We hope our clarification is helpful and we look forward to further discussions on these matters.

My biggest concern is the main focus of this study. While the author highlighted previous studies to justify the necessity of introducing CMCD into studying the dynamic process of NMC during operation, I can only find one set of data collected from CMCD, which is shown in Figure 1. In fact, most of the results that provide direct information on the local dynamic behavior come from optical images and subsequent analysis presented in the rest of the manuscript. Therefore, while the author states that "From a technical perspective, we highlight that the herein demonstrated CMCD approach is a strong extension to the conventional XRD method," it is not convincing.

CMCD is relatively new to the field of battery community. To our knowledge, there are very few papers utilizing CMCD for battery research. In comparison to traditional powder XRD techniques, CMCD offers advantages by probing the local structure with domain resolving power, which is important for understanding the domain behavior in lithium-ion batteries.

We would point out that the current implementation of CMCD has its limitations. For example, when significant morphological change occurs, it becomes very difficult to track the diffraction spots throughout the entire experiment. The physical translation, rotation, and deformation of active particles as well as electrodes can be induced by a variety of chemical and physical processes during the cell operation. This is highly relevant to the battery degradation and failure analysis. This challenge could

potentially be addressed using an X-ray source with a higher level of coherence. The implementation of simultaneous sample rocking and scanning is another viable approach to facilitate the tracking of diffraction spots with an improved robustness against the sample deformation.

We have included some of these discussions in the revised supplementary materials.

The current implementation of CMCD has its limitations. For example, when significant morphological change occurs, it becomes very difficult to track the diffraction spots throughout the entire experiment. The physical translation, rotation, and deformation of active particles and electrodes can be induced by a variety of chemical and physical processes during the cell operation. This challenge could potentially be addressed through the implementation of simultaneous sample rocking and scanning, for which a synchrotron source with higher brilliance and coherence is desirable.

Additionally, from my own perspective, CMCD is still an averaging X-ray diffraction-based technique as it does not provide spatial resolution and is unable to resolve local information in real space. This is why the authors further employed operando optical microscopy techniques to observe the local information of a single NMC particle. I wonder what new information exactly the CMCD technique provided if the optical microscope can already fulfill the requirements?

We thank Reviewer 4# for the insights into advanced X-ray techniques. It's important to note that while CMCD technology may not directly provide local information in real space, it serves as a valuable tool for observing the asynchrony among primary particles.

The monitoring of each diffraction spot separately can offer very useful insights. Specifically, the directions of the peak motion have different implications. As illustrated in Figure 27 (refer to Figure 1g), the peak trajectory vector can be divided into two independent components: 1) being normal to the ring, changing 2θ and 2) being along the tangential direction, changing η . The first component is relevant to the lattice breathing and the second component is resulted from the physical rotation of the grain. Therefore, the domain-level chemistry and the real-space motion of the primary grains can be inferred by analyzing the trajectory of the diffraction peak.

Figure R27: Schematic illustration of CMCD pattern depicting lattice contraction and domain rotation in the operating NMC lithium-ion cell (Figure 1g in the current paper)

In our view, the CMCD technique and the microscopy tools are highly complementary. For example, the limited penetrating capability of visible light restricts us to study particles exposed at the cross section. In contrast, CMCD looks into the buried primary domains and corroborates our optical observation. Furthermore, as a diffraction technique, CMCD offers insights into the domains lattice evolution, which is not directly accessible with the optical microscopy.

Most importantly, the main finding of this paper has been reported with similar results and techniques using an optical microscope. For instance, Xu et al. (2022) observed kinetically induced lithium heterogeneities in single-particle layered Ni-rich cathodes through operando visualization. I suggest that the authors provide a discussion differentiating the new findings in their manuscript.

We have carefully reviewed and compared this work (ref.#18 in our manuscript) with our paper. While both studies involve similar research subjects and experimental methods, it's crucial to highlight that our findings are distinct.

Firstly, our study focuses on a more complex system involving polycrystalline particles, which introduces additional complexities compared to the research cited, which concerns one single-crystal particle. We emphasize the direct charge transfer between primary domains within individual particles, which is a central aspect of our investigation.

Moreover, our work emphasizes the significance of statistical analysis. We conducted an in-depth analysis of over 100 particles, which provides a robust and comprehensive understanding for our study. This extensive dataset allows us to draw robust conclusions on the domain dynamics.

Additionally, we combined CMCD and optical microscopy to facilitate a cross exam. We leverage the strengths of both techniques to study the evolution of local SOC and lattice dynamics. This multi-modal approach offers a comprehensive understanding of the dynamic processes taking place in the battery cathode materials.

Finally, we would like to highlight that with observation of over 100 particles, core-shell model is invalidated in our system. Instead, a reaction front propagation pattern has been identified, as depicted in Figure R28. In contrast, a core-shell pattern was observed in the work by Xu et al (as shown in figure R29). This difference could arise due to the particle formation (single-crystal versus poly-crystal), which is a very interesting observation that warrants follow-up studies.

Figure R28 (a) Images of a typical NMC particle acquired using an optical microscope during the first charge. (b) Sequential differential maps reveal the sub-particle active regions with high local current. (Figure 2 a and b in the manuscript of the current paper)

Figure R29 Summary of the lithium-ion distribution within the single active particle at various lithium contents. Figure is reused from (DOI:10.1016/j.joule. 2022.09.008).

We appreciate the opportunity to clarify these differences and to highlight the strengths of our research.

REVIEWERS' COMMENTS

Reviewer #2 (Remarks to the Author):

The authors have addressed my previous concerns. I have no further comment.

Reviewer #3 (Remarks to the Author):

I acknowledge that the authors did a thorough work in the revision and, although they did not change much in their manuscript, they provided sound replies to most queries of myself and the other reviewers.

I am still however a little skeptical. The novelty of the paper as also highlighted by rev 4 is not quite there. Some nuances are new, but not the core I would say.

However I would see this more as an editorial decision.

There is also one scientific aspect that I find troubling, related to the comment reported below. I do not find the explanation of the authors convincing on this point. While I understand the heterogeneity and asynchronicity, I have a hard time believing a particle discharges while the others are still charging. there isn't only the gradient in Li concentration that may (perhaps) drive this, but also an externally imposed current that would prevent reduction of any particle which is electrically connected.

In my opinion this is still a major open point, which could even be indicative of some issue in the data processing or at the level of cell setup (things such as pressure homogeneity etc).

- I really don't understand how it is physically possible that a particle domain can discharge while the externally imposed current demands charge. This is totally counter intuitive, not to say unphysical. "D5 initiates its charging process at the beginning of the cell charge and reaches to its maximum SOC well before the end of the cell charge. Interestingly, while the whole cell is still being charged between time stamps 3 and 4, D5 already reverse its trend and starts to demonstrate a local discharging behavior. The authors should spend a lot more time on this point and try to propose a physically meaningful explanation.

The phenomenon of a particle domain being discharged while the overall cell is being charged is indeed intriguing and may seem counterintuitive at first glance. However, we can elaborate on this point to provide a meaningful explanation. The charging or discharging of a local domain within the cell can be understood as the diffusion of lithium ions from regions with higher lithium-ion concentration to regions

with lower lithium-ion concentration. This microscopic behavior could deviate from the macroscopic observation at some local regions.

For example, even though the entire cell is in being discharged, the local state of charge over a particular domain could be significantly higher or lower than the cell SOC due to the interplay of the asynchronicity and heterogeneity (as we observed and elaborated in this work). Therefore, lithium-ion can still diffuse from low SOC domains to high SOC domains, effectively causing the local charging behavior. The cell-level electrochemical signals (e.g., the cell current and voltage) are an accumulated effect of all the active particles in the system. Therefore, discrepancies can still occur as outliers in the system.

Similar phenomenon has also been observed in other electrochemical systems. For example, in *Nat. Nanotechnol.* 18, 790–797 (2023) (as shown in Figure R21), non-monotonic local dissolution/deposition behavior was observed in an aqueous battery system, which is another good example of the interplay between asynchronicity and heterogeneity in chemical systems.

Reviewer #4 (Remarks to the Author):

The authors have tried their best to answer my question. I agree that the main focus of this manuscript is on the phenomenon between multiple grains, which is different from the previous studies. However, I am not convinced that CMCD plays a significant role in this manuscript.

Point-to-point revision summary for

Article NCOMMS-23-36968A: *Asynchronous Domain Dynamics and Equilibration in Layered Oxide Battery Cathode*

Reviewer comments are color-coded in black

Author responses are color-coded in blue

Manuscript revisions are color-coded in red

We would like to express our sincere gratitude to all the reviewers for their efforts in assessing the manuscript we submitted and for their invaluable input during the review process. We are particularly pleased with the recognition given by three reviewers to the modifications made in the previous round. Our thanks extend to Reviewers 2 and 4 for their positive comments on the current state of the article. We are grateful for the scientific rigor that Reviewer 3 has applied. We deeply value the review process as each interaction with the editors and reviewers has been both enjoyable and highly beneficial. These engagements have not only enhanced the quality of our manuscript but also enriched our understanding of the scientific issues at hand.

Below, we have summarized our point-by-point response to the comments and suggestions made. We look forward to further interactions with all the reviewers and the editorial team in the future, confident that our collective efforts will result in more publications of the highest quality.

Reviewer #2 (Remarks to the Author):

The authors have addressed my previous concerns. I have no further comment.

We extend our sincere thanks for your constructive feedback and are pleased to have addressed all of their previous concerns satisfactorily.

Reviewer #3 (Remarks to the Author):

I acknowledge that the authors did a thorough work in the revision and, although they did not change much in their manuscript, they provided sound replies to most queries of myself and the other reviewers.

I am still however a little skeptical. The novelty of the paper as also highlighted by rev 4 is not quite there. Some nuances are new, but not the core I would say.

However I would see this more as an editorial decision.

Thank you for your insightful comments. We would like to highlight that our study focuses on the dynamics of polycrystalline NMC, an area that is critical to the understanding of lithium-ion battery performance. The dynamics within these materials are essential, and while Reviewer 4 has pointed to research on single-crystal NMC, our work distinctively investigates the multifaceted dynamics of multiple domains in polycrystalline NMC, which adds a significant layer of complexity and relevance.

There is also one scientific aspect that I find troubling, related to the comment reported below. I do not find the explanation of the authors convincing on this point. While I understand the heterogeneity and asynchronicity, I have a hard time believing a particle discharges while the others are still charging. there isn't only the gradient in Li concentration that may (perhaps) drive this, but also an externally imposed current that would prevent reduction of any particle which is electrically connected.

In my opinion this is still a major open point, which could even be indicative of some issue in the data processing or at the level of cell setup (things such as pressure homogeneity etc).

- I really don't understand how it is physically possible that a particle domain can discharge while the externally imposed current demands charge. This is totally counter intuitive, not to say unphysical. "D5 initiates its charging process at the beginning of the cell charge and reaches to its maximum SOC well before the end of the cell charge. Interestingly, while the whole cell is still being charged between time stamps 3 and 4, D5 already reverse its trend and starts to demonstrate a local discharging behavior. The authors should spend a lot more time on this point and try to propose a physically meaningful explanation.

The observed phenomenon of a particle domain charging while the entire cell is discharging is indeed intriguing. The discharging of the whole battery system is driven by the external circuit's current, which on an average level, initiates discharging. However, due to asynchrony and heterogeneity within the battery, local domains exhibit significant differences in SOC. In the scenario presented in our paper, even though domain D6 has begun lithiation, the lag in charging of domain D5 means that the lithium concentration in D6 could still be considerably lower than in D5, allowing lithium ions to be driven from D5 to D6 by the concentration gradient. This diffusion could be investigate as local charging. This is consistent with phenomena we have observed in our CMCD results.

In this figure, although the whole cell is under a discharging current, some primary particles are still in the process of charging, trying to reach equilibrium due to a lag in

the charging process. These charging particles are effectively 'catching up' with the rest of the cell.

Reviewer #4 (Remarks to the Author):

The authors have tried their best to answer my question. I agree that the main focus of this manuscript is on the phenomenon between multiple grains, which is different from the previous studies. However, I am not convinced that CMCD plays a significant role in this manuscript.

Thank you for your continued engagement with our work. We appreciate your perspective and would like to emphasize that our research is centered on the behavior of domains in the layered oxide cathodes of lithium-ion batteries. The CMCD is indeed an analytical tool we've utilized to support our investigation, not the primary focus. We hope this clarification aligns with your understanding.